# RANDOM MASK: TOWARDS ROBUST CONVOLUTIONAL NEURAL NETWORKS

**Tiange Luo**[1]*, **Tianle Cai**[1]*, **Mengxiao Zhang**[2], **Siyu Chen**[1], **Liwei Wang**[1]
[1]Peking University, [2]University of Southern California
[1]{luotg,caitianle1998,siyuchen,wanglw}@pku.edu.cn, [2]zhan147@usc.edu

## ABSTRACT

Robustness of neural networks has recently been highlighted by the *adversarial examples*, i.e., inputs added with well-designed perturbations which are imperceptible to humans but can cause the network to give incorrect outputs. In this paper, we design a new CNN architecture that by itself has good robustness. We introduce a simple but powerful technique, *Random Mask*, to modify existing CNN structures. We show that CNNs with Random Mask achieve state-of-the-art performance against black-box adversarial attacks *without* applying any adversarial training. We next investigate the adversarial examples which "fool" a CNN with Random Mask. Surprisingly, we find that these adversarial examples often "fool" humans as well. This raises fundamental questions on how to define adversarial examples and robustness properly.

## 1 INTRODUCTION

Deep learning (LeCun et al., 2015), especially deep Convolutional Neural Network (CNN) (LeCun et al., 1998), has led to state-of-the-art results spanning many machine learning fields, such as image classification (He et al., 2016; Hu et al., 2017b; Huang et al., 2017; Simonyan & Zisserman, 2014), object detection (Redmon et al., 2016; Girshick, 2015; Ren et al., 2015), image captioning (Vinyals et al., 2015; Xu et al., 2015) and speech recognition (Bengio et al., 2003; Hinton et al., 2012).

Despite the great success in numerous applications, recent studies have found that deep CNNs are vulnerable to some *well-designed* input samples named as *Adversarial Examples* (Szegedy et al., 2013) (Biggio et al., 2013). Take the task of image classification as an example, for almost every commonly used well-performed CNN, attackers are able to construct a small perturbation on an input image to cause the model to give an incorrect output label. Meanwhile, the perturbation is almost *imperceptible to humans*. Furthermore, these adversarial examples can easily *transfer* among different kinds of CNN architectures (Papernot et al., 2016b).

Such adversarial examples raise serious concerns on deep neural network models as robustness is crucial in many applications. Just as Goodfellow (2018) suggests, both robustness and traditional supervised learning seem fully aligned. Recently, there is a rapidly growing body of work on this topic. One important line of research is adversarial training (Szegedy et al., 2013; Madry et al., 2017; Goodfellow et al., 2015; Huang et al., 2015). Although adversarial training gains some success, a major difficulty is that it tends to overfit to the method of adversarial example generation used at training time (Buckman et al., 2018). Xie et al. (2017) and Guo et al. (2017) propose defense methods by introducing randomness and applying transformations to the inputs respectively. Dhillon et al. (2018) introduces random drop during the *evaluation* of a neural network. However, Athalye et al. (2018) contends that such transformation and randomness only provide a kind of "obfuscated gradient" and can be attacked by taking expectation over transformation (EOT) to

---

*Equal contribution.

get a meaningful gradient. Papernot et al. (2016a) and Katz et al. (2017) consider the non-linear functions in the networks and try to achieve robustness by adjusting them. There are also detection-based defense methods (Metzen et al., 2017; Grosse et al., 2017; Meng & Chen, 2017), which add a process of detecting whether an input is adversarial.

In this paper, different from most of the existing methods, we take another approach to tackle the adversarial example problem. In particular, we aim to design a new CNN architecture which by itself enjoys robustness, without appealing to techniques such as adversarial training. To this end, we introduce Random Mask as a new ingredient of CNN. To be specific, we randomly select a set of neurons and remove them from the network *before* training. Then the architecture of the network is *fixed* during the training and testing process. Note that Random Mask is different from dropout which randomly masks out neurons in each step *during* training. In addition, Random Mask can be applied very easily to common CNN structures such as ResNet with only a few changes of code. We find that applying Random Mask to the shallow layers of the network is crucial for robustness. CNNs with properly designed Random Mask are far more robust than standard CNNs. In fact, our experimental results demonstrate that CNNs with Random Mask achieve state-of-the-art results against black-box attacks even when comparing with defense methods using adversarial training. Furthermore, CNNs with Random Mask maintain a high accuracy on normal test data, while low test accuracy is often regarded as a major weakness in many methods designed for achieving robustness.

We next take a closer look at the adversarial examples generated particularly against CNNs with Random Mask. We investigate the adversarial examples that can "fool" our proposed architecture, i.e., the examples that are perturbed version of the original image, but are classified to a different label by the network. Surprisingly, we find that the adversarial examples which can "fool" a CNN with Random Mask often "fool" humans as well. It is difficult for humans to correctly classify these adversarial example, and in many cases humans make the same "incorrect" prediction as our network. Figure 1 shows a few adversarial examples generated by PGD (Basic iterative method) (Kurakin et al., 2016) with respect to a CNN with Random Mask as well as the labels the network outputs. (Please also see Figure 12 in Appendix E.1 for the original images and labels from CIFAR-10.) They are different from the typical adversarial examples generated against commonly used CNNs, which usually look like noisy versions of the original images and are easy to be correctly classified by humans.

These observations raise important questions: 1) How should we define adversarial examples? 2) How should we define robustness? Currently, an adversarial example is usually defined as a perturbed datum which lies in the neighborhood of the original data but has a different classification output by the network; and the robustness of a method is measured according to the proportion of these adversarial examples. However, if an adversarial example can also fool humans, it is more appropriate to say that the example does change the semantics of the datum to a certain extent. After all, why two images close to each other in terms of some distance (e.g., $\ell_\infty$) must belong to the same class? How close should they be so that they belong to the same class? Without complete answers to these questions, one should be very careful when measuring the robustness of a model merely according to currently-defined adversarial examples. Robustness is a subtle issue. We argue that one needs to rethink the robustness and adversarial examples from the definitions.

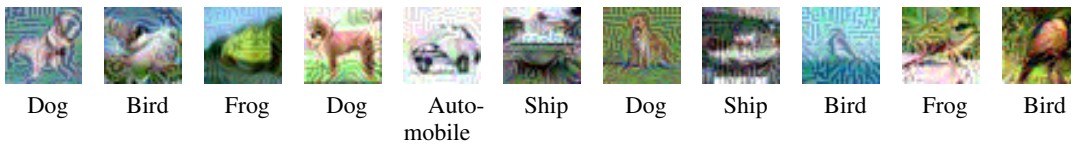

| Dog | Bird | Frog | Dog | Auto-mobile | Ship | Dog | Ship | Bird | Frog | Bird |

Figure 1: Adversarial examples (generated by PGD against a network with Random Mask) that can "fool" a CNN with Random Mask. The labels here are the outputs of the network being "fooled". The original images from CIFAR-10 and more examples can be found in Figure 12 in Appendix E.1.

Our main contributions are summarized as follows:

- We develop a very simple but effective method, *Random Mask*. We show that combining with Random Mask, existing CNNs can be significantly more robust while maintaining high generalization performance. In fact, CNNs equipped with Random Mask achieve state-of-the-art performance against several black-box attacks, even when comparing with methods using adversarial training (See Table 1).

- We investigate the adversarial examples generated against CNNs with Random Mask. We find that adversarial examples that can "fool" a CNN with Random Mask often fool humans as well. This observation requires us to rethink what are the right definitions of adversarial examples and robustness.

## 2  RANDOM MASK

We propose Random Mask, a method to modify existing CNN structures. It randomly selects a set of neurons and removes them from the network *before* training. Then the architecture of the network is *fixed* during the training and testing process. To apply Random Mask on a selected layer $Layer(j)$, suppose the input is $X_j$ and the output is $conv_j(X_j) \in \mathbb{R}^{m_j \times n_j \times c_j}$. We randomly generate a binary mask $mask(j) \in \{0,1\}^{m_j \times n_j \times c_j}$ by sampling uniformly within each channel. The *drop rate* of the sampling process is called the *ratio* (or *drop ratio*) of Random Mask. Then we mask the neurons in position $(x, y, c)$ of the output of $Layer(j)$ if the $(x, y, c)$ element of $mask(j)$ is zero. More specifically, after Random Mask, we will not compute these masked neurons and make the next layer regard these neurons as having value zero during computation. A simple visualization of Random Mask is shown in Figure 2. The Random Mask in fact decreases the computational cost in each epoch since there are fewer effective connections. Note that the *number of parameters* in the convolutional kernels remains *unchanged*, since we only mask neurons in the feature maps.

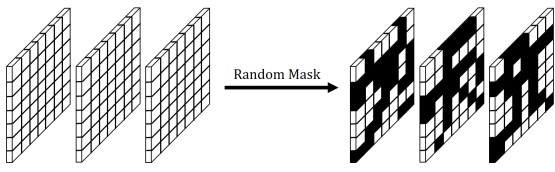

Figure 2: An illustration of Random Mask applied to three *channels* of a layer (neuron-wise). Note that the number of parameters in the network is *not reduced* after applying Random Mask.

In the standard setting, convolutional filter will be applied uniformly to every position of the feature map of the former layer. The success of this implementation is due to the reasonable assumption that if one feature is useful to be computed at some spatial position $(x, y)$, then it should also be useful to be computed at a different position $(x', y')$. Thus the original structure is powerful for feature extraction. Moreover, this structure leads to parameter sharing which makes the training process more efficient. However, the uniform application of filter also prevents the CNN from noticing the distribution of features. In other words, the network focuses on the *existence* of a kind of feature but pays little attention to how this kind of feature *distributes on the whole feature map* (of the former layer). Yet the pattern of feature distribution is important for humans to recognize and classify a photo, since empirically people would rely on some structured feature to perform classification.

With Random Mask, each filter may only extract features from partial positions. More specifically, for one filter, only features which distribute consistently with the mask pattern can be extracted. Hence filters in a network with Random Mask may capture more information on the spatial structures of local features. Just think of a toy example: imagine Random Mask for a filter masks all the neurons but one row in the channel,

if a kind of feature usually distributes in a column, it can not have strong response because the filter can only capture a small portion of the feature.

We do a straightforward experiment to verify our intuition. We sample some images from ImageNet which can be correctly classified with high probability by both CNNs with and without Random Mask. We then randomly shuffle the images by patches, and compare the accuracy of classifying the shuffled images (See Appendix A). We find out that the accuracy of the CNN with Random Mask is consistently lower than that of normal CNN. This result shows that CNNs without Random Mask cares more about *whether a feature exists* while CNNs with Random Mask will *detect spatial structures* and *limit poorly-organized features from being extracted*.

We further explore how Random Mask plays its role in defending against adversarial examples. Recent observation (Liu et al., 2018) of adversarial examples found that these examples usually change a patch of the original image so that the perturbed patch looks like a small part of the incorrectly classified object. This perturbed patch, although contains crucial features of the incorrectly classified object, usually appears at the wrong location and does not have the right spatial structure with other parts of the image. For example (See Figure 11 in Liu et al. (2018)), the adversarial example of a panda image is misclassified as a monkey because a patch of the panda skin is perturbed adversarially so that it alone looks like the monkey's face. However, this patch does not form a right structure of a monkey with other parts of the images. By the properties of detecting spatial structures and limiting feature extraction, Random Mask can naturally help CNNs resist such adversarial perturbations.

In complement to the observation we mentioned above, we also find that most adversarial perturbations generated against normal CNNs look like random noises which do not change the semantic information of the original image. In contrast, adversarial examples generated against CNNs with Random Mask tend to contain some well-organized features which sometimes change the classification results semantically (See Figure 3 and Figure 1). This phenomenon also supports our intuition that Random Mask helps to detect spatial structures and extract well-organized features via imposing limitations.

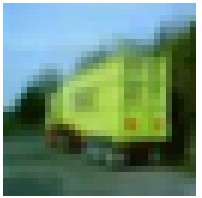 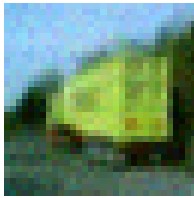 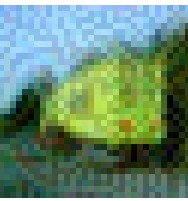 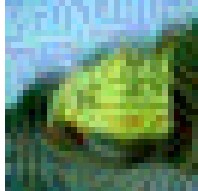

Original Image      Gaussian Noise      Normal CNN      Random Mask

Figure 3: The first image is the original image, and the other three contain different types of small perturbations. Both the two adversarial examples on the right are predicted as frog by the corresponding models. However, only the image generated by the randomly masked CNN is capable of fooling humans.

While the features that can be learned by each masked filter is limited, the *randomness* helps us get plenty of diversified patterns. Our experiments show that these limited filters are enough for learning features. In other words, CNNs will maintain a high test accuracy after being applied with Random Mask. Besides, adding convolutional filters may help our CNN with Random Mask to increase test accuracy (See Section 3.3). Furthermore, our structure is naturally compatible to ensemble methods, and randomness makes ensemble more powerful (See Section 3.3).

However, it might not be appropriate to apply Random Mask to deep layers. The distribution of features is meaningful only when the location in feature map is *highly related to the location in the original input image*,

and the receptive field of each neuron in deep layers is too large. In Section 3.3, there are empirical results which support our intuition.

## 3 EXPERIMENTS

In this section, we provide extensive experimental analyses on the performance and properties of *Random Mask* network structure. We first test the robustness of Random Mask (See Section 3.1). Then we take a closer look at the adversarial examples that can "fool" our proposed architecture (See Section 3.2). After that we explore properties of Random Mask, including where and how to apply Random Mask, by a series of comparative experiments (See Section 3.3). Some settings used in our experiments are listed below:

**Network Structure.** We apply Random Mask to several target networks, including ResNet-18 (He et al., 2016), ResNet-50, DenseNet-121 (Huang et al., 2017), SENet-18 (Hu et al., 2017b) and VGG-19 (Simonyan & Zisserman, 2014). The effects of Random Mask on those network structures are quite consistent. For brevity, we only show the defense performance on ResNet-18 in the main body and leave more experimental results in the Appendix F. The 5-block structure of ResNet-18 is shown in the Appendix C.1. The blocks are labeled $0, 1, 2, 3, 4$ and the $0^{th}$ block is the first convolution layer. We divide these five blocks into two parts - the relative shallow ones (the $0^{th}, 1^{st}, 2^{nd}$ blocks) and the deep ones (the $3^{rd}, 4^{th}$ blocks). For simplicity, we would like to regard each of these two parts as a whole in this section to avoid being trapped by details. We use "$\sigma$-Shallow" and "$\sigma$-Deep" to denote that we apply Random Mask with drop ratio $\sigma$ to the shallow blocks and to the deep blocks in ResNet-18 respectively.

**Attack Framework.** The *accuracy under black-box attack* serves as a common criterion of robustness. We will use it when selecting model parameters and comparing Random Mask to other similar structures. To be more specific, by using FGSM (Goodfellow et al., 2015), PGD (Kurakin et al., 2016) with $\ell_\infty$ norm and CW attack (Carlini & Wagner, 2016) with $\ell_2$ norm (See Appendix B for details on these attack approaches), we generate adversarial examples against different neural networks. The performances on adversarial examples generated against different networks are quite consistent. For brevity, we only show the defense performance against part of the adversarial examples generated by using DenseNet-121 on dataset CIFAR-10 in this section, and leave more experimental results obtained by using other adversarial examples in the Appendix F. We use **FGSM**$_{16}$, **PGD**$_{16}$, **PGD**$_{32}$, **CW**$_{40}$ to denote attack method FGSM with step size $\epsilon = 16$, PGD with perturbation scale $\alpha = 16$ and step number 20, PGD with perturbation scale $\alpha = 32$ and step number 40, CW attack with confidence $\kappa = 40$ respectively. The step size of both PGD methods are selected to be $\epsilon = 1$. We would like to point out that these attacks are really powerful that a normal network cannot resist these attacks.

### 3.1 ROBUSTNESS VIA RANDOM MASK

Random Mask is not specially designed for adversarial defense, but as Random Mask introduces information that is essential for classifying correctly, it also brings robustness. As mentioned in Section 2, normal CNN structures may allow adversary to *inject features imperceptible to humans* into images that can be recognized by CNN. Yet Random Mask limits the process of feature extraction, so noisy features are less likely to be preserved.

### 3.1.1 ROBUSTNESS TO BLACK-BOX ATTACK

The results of our experiments show the strengths of applying Random Mask to adversarial defense. In fact, Random Mask can help existing CNNs reach state-of-the-art performance against the black-box attacks we use (See Table 1). In Section 3.3, we will provide more experimental results to show that this asymmetric structure performs better than normal convolution and enhances robustness.

| Model | FGSM | PGD | Test Accuracy |
|---|---|---|---|
| Normal ResNet-18 | 26.99% | 7.56% | 95.33% |
| Vanilla (Madry) | 85.60% | 86.00% | 87.30% |
| Random Mask | **86.31%** | **90.30%** | **90.08%** |

Table 1: Performance of black-box defense under the setting of Madry et al. (2017) (See Appendix F.1 for the complete setting). We use model under adversarial training in Madry et al. (2017) as a vanilla model. It is regarded as a state-of-the-art adversarial defense method. Our model is only trained on *clean data*. The ratio of Random Mask here is selected to balance the performance of robustness and generalization. See Figure 14 in Appendix F.1 for results on the performance of Random Mask with different ratios.

### 3.1.2 ROBUSTNESS TO RANDOM NOISE

Beside the robustness against black-box attack, we also evaluate the robustness to random noise. Note that although traditional network structures are vulnerable to adversarial examples, they are still robust to the images perturbed with small Gaussian noises. To see whether our structure also enjoys such property, or even has better robustness in this sense, we feed input images with random Gaussian noises to networks with Random Mask. More specifically, in order to obtain noises of scales similar to the adversarial perturbations, we generate i.i.d. Gaussian random variables $x \sim N(0, \sigma^2)$, where $\sigma \in \{1, 2, 4, 8, 12, 16, 20, 24, 28, 32\}$, clip them to the range $[-2\sigma, 2\sigma]$ and then add them to every pixel of the input image. The results of the experiments are shown in Figure 4. We can see that networks with Random Mask always have higher accuracy than a normal network.

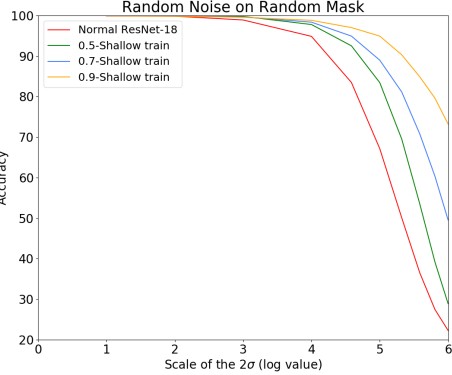

Figure 4: Input images with random Gaussian noises, Random Mask versus normal network. The network with Random Mask has far better robustness than the original network.

### 3.2 ADVERSARIAL EXAMPLES?

We evaluate the performance of CNNs with Random Mask under white-box attack (See Appendix F.3). With neither obfuscated gradient nor gradient masking, Random Mask can still improve defense performance under various kinds of white-box attack. Also, by checking adversarial images that are misclassified by our network, we find most of them have vague edges and *can hardly be recognized by humans*. This result coincides with the theoretical analysis in Shafahi et al. (2018); Fawzi et al. (2018) that real adversarial examples may be

inevitable in some way. See Appendix E.2 for a randomly selected set of them. In contrast, adversarial examples generated against normal CNNs are more like simply adding some non-sense noise which can be ignored by human. This phenomenon also demonstrates that Random Mask really helps networks to catch more information related to real human perception. Moreover, just as Figure 1 shows, with the help of Random Mask, we are able to find small perturbations that can actually change the semantic meaning of images for humans. So should we still call them "adversarial examples"? How can we get more reasonable definitions of adversarial examples and robustness? These questions seem severe due to our findings.

### 3.3 PROPERTIES OF RANDOM MASK

We then show some properties of Random Mask including the appropriate positions to apply Random Mask, the benefit of breaking symmetry, the diversity introduced by randomness and the extensibility of Random Mask via structure adjustment and ensemble methods. We conduct a series of comparative experiments and we will continue to use black-box defense performance as a criterion of robustness. For brevity, we only present the results of a subset of our experiments in Table 2. Full information on all the experiments can be found in Appendix F.5.

| Network Structure | $FGSM_{16}$ | $PGD_{16}$ | $PGD_{32}$ | $CW_{40}$ | Test Accuracy |
|---|---|---|---|---|---|
| Normal ResNet-18 | 14.91% | 2.96% | 2.26% | 8.23% | 95.33% |
| 0.3-Shallow | 23.29% | 14.53% | 5.73% | 36.95% | 94.03% |
| 0.5-Shallow | 30.86% | 26.50% | 10.33% | 54.02% | 93.39% |
| 0.7-Shallow | 48.57% | 47.76% | 21.39% | 73.70% | 91.83% |
| 0.3-Deep | 14.62% | 1.95% | 1.30% | 7.88% | 95.16% |
| 0.5-Deep | 10.76% | 2.57% | 4.52% | 7.19% | 94.94% |
| 0.7-Deep | 11.23% | 3.24% | 2.64% | 10.10% | 94.61% |
| 0.3-Shallow, 0.3-Deep | 24.15% | 12.67% | 6.75% | 29.65% | 94.16% |
| 0.3-Shallow, 0.7-Deep | 11.26% | 7.94% | 5.77% | 23.31% | 93.44% |
| 0.7-Shallow, 0.7-Deep | 27.43% | 32.72% | 16.26% | 62.47% | 89.78% |
| 0.7-Shallow, 0.3-Deep | 40.58% | 42.95% | 19.00% | 68.58% | 91.23% |
| 0.5-Shallow | 30.86% | 26.50% | 10.33% | 54.02% | 93.39% |
| 0.9-Shallow | 79.93% | 83.08% | 55.02% | 89.67% | 87.68% |
| $0.5\text{-Shallow}_{DC}$ | 12.15% | 4.68% | 4.05% | 12.72% | 94.97% |
| $0.9\text{-Shallow}_{DC}$ | 19.00% | 19.33% | 10.08% | 44.80% | 93.27% |
| $0.5\text{-Shallow}_{SM}$ | 48.86% | 44.04% | 19.81% | 72.07% | 92.57% |
| $0.9\text{-Shallow}_{SM}$ | 39.40% | 50.40% | 29.23% | 65.38% | 74.28% |
| $0.5\text{-Shallow}_{\times 2}$ | 20.78% | 12.51% | 5.38% | 34.00% | 94.12% |
| $0.9\text{-Shallow}_{\times 2}$ | 68.83% | 66.86% | 37.51% | 82.74% | 90.49% |
| $0.9\text{-Shallow}_{\times 4}$ | 59.64% | 59.15% | 32.29% | 78.88% | 90.57% |
| $\text{Normal ResNet-18}_{EN}$ | 16.24% | 2.22% | 1.46% | 8.58% | 96.12% |
| $0.5\text{-Shallow}_{\times 2,EN}$ | 19.84% | 11.86% | 5.30% | 37.37% | 95.24% |
| $0.5\text{-Shallow}_{EN}$ | 31.38% | 27.58% | 9.97% | 58.07% | 94.56% |
| $0.9\text{-Shallow}_{EN}$ | 81.95% | 85.14% | 56.02% | 91.36% | 89.45% |

Table 2: A subset of our experiments presented in Appendix F.5 to show properties of Random Mask. $\sigma\text{-Shallow}_{DC}$, $\sigma\text{-Shallow}_{SM}$, $\sigma\text{-Shallow}_{\times n}$ and $\sigma\text{-Shallow}_{EN}$ mean dropping channels with ratio $\sigma$, applying same mask with ratio $\sigma$, increasing channel number to $n$ times with mask ratio $\sigma$ for every channel and ensemble five models with different masks of same ratio $\sigma$ respectively. The entries in the middle four columns are success rates of defense under different settings. This is also .

**Masking Shallow Layers versus Masking Deep Layers.** In the last paragraph of Section 2 , we give an intuition that deep layers in a network should not be masked. To verify this, we do extensive experiments on ResNet-18 with Random Mask applied to different parts. We apply Random Mask with different ratios on the shallow blocks and on the deep blocks respectively. Results in Table 2 accord closely with our intuition. Comparing the success rate of black-box attacks on the model with the same drop ratio but different parts being masked, we find that applying Random Mask to *shallow layers* enjoys significantly lower adversarial attack success rates. This verifies that shallow layers play a more important role in limiting feature extraction than the deep layers. Moreover, only applying Random Mask on shallow blocks can achieve better performance than applying Random Mask on both shallow and deep blocks, which also verifies our intuition that dropping elements with large receptive fields is not beneficial for the network. In addition, we would like to point out that ResNet-18 with Random Mask significantly *outperforms* the normal network in terms of *robustness*.

**Random Mask versus Channel Mask.** As our Random Mask applies independent random masks to different channels in a layer, we actually break the symmetry of the original CNN structure. To see whether this asymmetric structure would help, we try to directly drop whole channels instead of neurons using the same drop ratio as the Random Mask and train it to see the performance. This channel mask does not hurt the symmetry while also leading to the same decrease in convolutional operations. Table 2 shows that although our Random Mask network suffers a small drop in test accuracy due to the high drop ratio, we have a great gain in the robustness, compared with the channel-masking network.

**Random Mask versus Same Mask.** The randomness in generating masks in different channels and layers allows each convolutional filter to focus on different patterns of feature distribution. We show the essentialness of generating various masks per layer via experiments that compare Random Mask to a method that only randomly generates one mask per layer and uses it in every channel. Table 2 shows that applying the same mask to each channel will decrease the test accuracy. This may result from the limitation of expressivity due to the monotone masks at every masked layer. In fact, we can illustrate such limitation using simple calculations. Since the filters in our base network ResNet-18 is of size $3 \times 3$, each element of the feature maps after the first convolutional layer can extract features from at most 9 pixels in the original image. This means that if we use the same mask and the drop ratio is $90\%$, only at most $9 \times 10\%$ of the input image can be caught by the convolutional layer, which would cause severe loss of input information.

**Increase the Number of Channels.** In order to compensate the loss of masking many neurons in each channel, it is reasonable that we may need more convolutional filters for feature extraction. Therefore, we try to increase the number of channels at masked layers. Table 2 shows that despite ResNet-18 is a well-designed network structure, increasing channels does help the network with Random Mask to get higher test accuracy while maintaining good robustness performance.

**Ensemble Methods.** Thanks to the diversity of Random Mask, we may directly use several networks with the same structure but different Random Masks and ensemble them. Table 2 shows that such ensemble methods can improve a network with Random Mask in both test accuracy and robustness.

## 4 CONCLUSION AND FUTURE DIRECTIONS

In conclusion, we introduce and experiment on Random Mask, a modification of existing CNNs that makes CNNs capture more information including the pattern of feature distribution. We show that CNNs with Random Mask can achieve much better robustness while maintaining high test accuracy. More specifically, by using Random Mask, we reach state-of-the-art performance in several black-box defense settings. Another insight resulting from our experiments is that the adversarial examples generated against CNNs with Random Mask actually change the semantic information of images and can even "fool" humans. We hope that this finding can inspire more people to rethink adversarial examples and the robustness of neural networks.

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

## A    RANDOM SHUFFLE

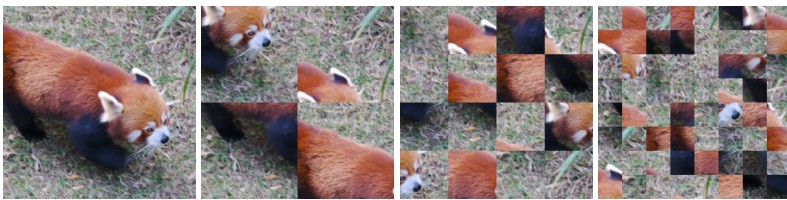

Figure 5: An example image that is randomly shuffled after being divided into $1 \times 1$, $2 \times 2$, $4 \times 4$ and $8 \times 8$ patches respectively.

In this part, we show results of our Random Shuffle experiment. Intuitively, by dropping randomly selected neurons in the neural network, we may let the network learn the relative margins and features better than normal networks. In randomly shuffled images, however, some global patterns of feature distributions are destroyed, so we expect that CNNs with Random Mask would have some trouble extracting feature information and might have worse performance than normal networks. In order to verify our intuition, we compare the test accuracy of a CNN with Random Mask to that of a normal CNN on randomly shuffled images. Specifically speaking, in the experiments, we first train a 0.7-Shallow network along with a normal network on ImageNet dataset. Then we select 5000 images from the validation set which are predicted correctly with more than 99% confidence by both normal and masked networks. We resize these images to $256 \times 256$ and then center crop them to $224 \times 224$. After that, we random shuffle them by dividing them into $k \times k$ small patches $k \in \{2, 4, 8\}$, and randomly rearranging the order of patches. Figure 5 shows one example of our test images after random shuffling. Finally, we feed these shuffled images to the networks and see their classification accuracy. The results are shown in Table 3.

| **Model** | $2 \times 2$ | $4 \times 4$ | $8 \times 8$ |
|---|---|---|---|
| Normal ResNet-18 | 99.58% | 82.66% | 17.56% |
| 0.7-Shallow | 97.36% | 64.00% | 11.94% |

Table 3: The accuracy by using normal and masked networks to classify randomly shuffled test images.

From the results, we can see that our network with Random Mask always has lower accuracy than the normal network on these randomly shuffled test images, which indeed accords with our intuition. By randomly shuffling the patches in images, we break the relative positions and margins of the objects and pose negative impact to the network with Random Mask since it may rely on such information to classify. Note that randomly shuffled images are surely difficult for humans to classify, so this experiment might also imply that the network with Random Mask is more similar to human perception than the normal one.

## B    ATTACK APPROACHES

We first give an overview of how to attack a neural network in some mathematical notations. Let $\boldsymbol{x}$ be the input to the neural network and $f_{\boldsymbol{\theta}}$ be the function which represents the neural network with parameter $\boldsymbol{\theta}$. The output label of the network to the input can be computed as $c = \arg\max_i f_{\boldsymbol{\theta}}(\boldsymbol{x})$. In order to perform an adversarial attack, we add a small perturbation $\delta_x$ to the original image and get an adversarial image $\boldsymbol{x}_{adv} = \boldsymbol{x} + \delta_x$. The new input $\boldsymbol{x}_{adv}$ should look visually similar to the original $\boldsymbol{x}$. Here we use the commonly

used $\ell_\infty$-norm metric to measure similarity, i.e., we require that $||\delta_x|| \leq \epsilon$. The attack is considered successful if the predicted label of the perturbed image $c_{adv} = \arg\max_i f_\theta(x_{adv})$ is different from $c$.

Generally speaking, there are two types of attack methods: *Targeted Attack*, which aims to change the output label of an image to a specific (and different) one, and *Untargeted Attack*, which only aims to change the output label and does not restrict which specific label the modified example should let the network output.

In this paper, we mainly use the following three attack approaches. $J$ denotes the loss function of the neural network and $y$ denotes the true label of $x$.

- **Fast Gradient Sign Method (FGSM).** FGSM (Goodfellow et al., 2015) is a one-step untargeted method which generates the adversarial example $x_{adv}$ by adding the sign of the gradients multiplied by a step size $\epsilon$ to the original benign image $x$. Note that FGSM controls the $\ell_\infty$-norm between the adversarial example and the original one by the parameter $\epsilon$.

$$x_{adv} = x + \epsilon \cdot \text{sign}(\nabla_x J(x, y)).$$

- **Basic iterative method (PGD).** PGD is a multiple-step attack method which applies FGSM multiple times. To make the adversarial example still stay "close" to the original image, the image is projected to the $\ell_\infty$-ball centered at the original image after every step. The radius of the $\ell_\infty$-ball is called perturbation scale and is denoted by $\alpha$.

$$x_{adv}^0 = x, \quad x_{adv}^{k+1} = Clip_{x,\alpha}\left[x_{adv}^k + \epsilon \cdot \text{sign}(\nabla_{x_{adv}^k} J(x_{adv}^k, y))\right].$$

- **CW Attack.** Carlini & Wagner (2016) shows that constructing an adversarial example can be formulated as solving the following optimization problem:

$$x_{adv} = \arg\min_{x'} c \cdot g(x') + ||x' - x||_2^2,$$

where $c \cdot g(x')$ is the loss function that evaluates the quality of $x'$ as an adversarial example and the term $||x' - x||_2^2$ controls the scale of the perturbation. More specifically, in the untargeted attack setting, the loss function $g(x)$ can be defined as:

$$g(x) = \max\{\max_{i \neq y}(f(x)_i) - f(x)_y, -\kappa\},$$

where the parameter $\kappa$ is called confidence.

## C  NETWORK ARCHITECTURES

Here we briefly introduce the network architectures used in our experiments. Generally, we apply Random Mask at the shallow layers of the networks and we have tried five different architectures, namely **ResNet-18**, **ResNet-50**, **DenseNet-121 SENet-18** and **VGG-19**. We next illustrate these architectures and show how we apply Random Mask to them.

### C.1  RESNET-18

ResNet-18 (He et al., 2016) contains 5 blocks: the $0^{th}$ block is one single $3 \times 3$ convolutional layer, and each of the rest contains four $3 \times 3$ convolutional layers. Figure 6 shows the whole structure of ResNet-18. In our experiment, applying Random Mask to a block means applying Random Mask to every layer in it.

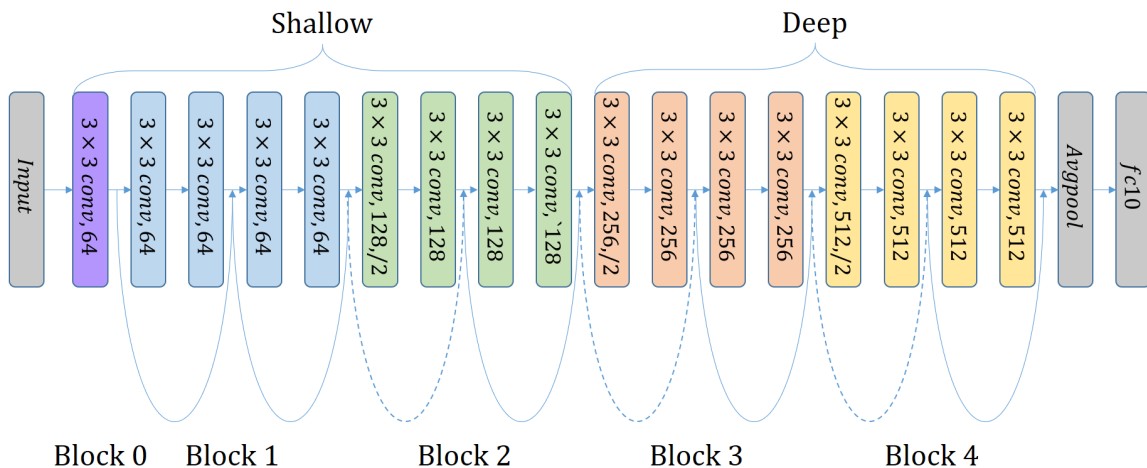

Figure 6: The architecture of ResNet-18

## C.2    RESNET-50

Similar to ResNet-18, ResNet-50 (He et al., 2016) contains 5 blocks and each block contains several $1 \times 1$ and $3 \times 3$ convolutional layers (i.e. Bottlenecks). In our experiment, we apply Random Mask to the $3 \times 3$ convolutional layers in the first three "shallow" blocks. The masked layers in the 1st block are marked by the red arrows.

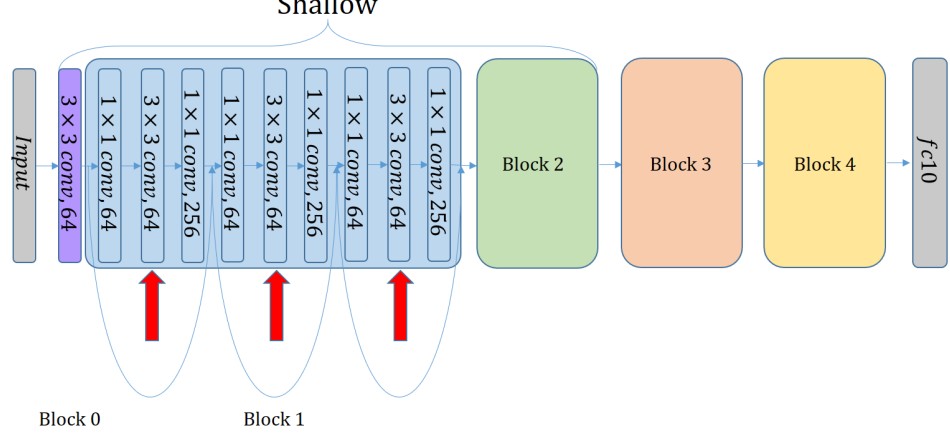

Figure 7: The architecture of ResNet-50

## C.3    DENSENET-121

DenseNet-121 (Huang et al., 2017) is another popular network architecture in deep learning researches. It contains 5 Dense-Blocks, each of which contains several $1 \times 1$ and $3 \times 3$ convolutional layers. Similar to what

we do for ResNet-50, we apply Random Mask to the $3 \times 3$ convolutional layers in the first three "shallow" blocks. The growth rate is set to 32 in our experiments.

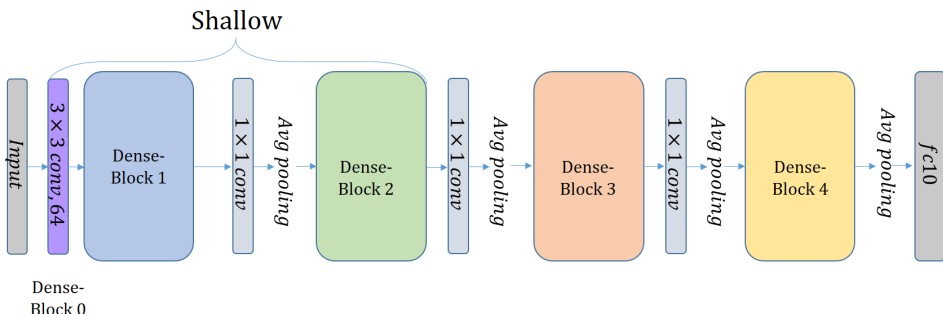

Figure 8: The architecture of DenseNet

## C.4 SENET

SENet (Hu et al., 2017a), a network architecture which won the first place in ImageNet contest 2017, is shown in Figure 9. Note that here we use the pre-activation shortcut version of SENet and we apply Random Mask to the convolutional layers in the first 3 SE-blocks.

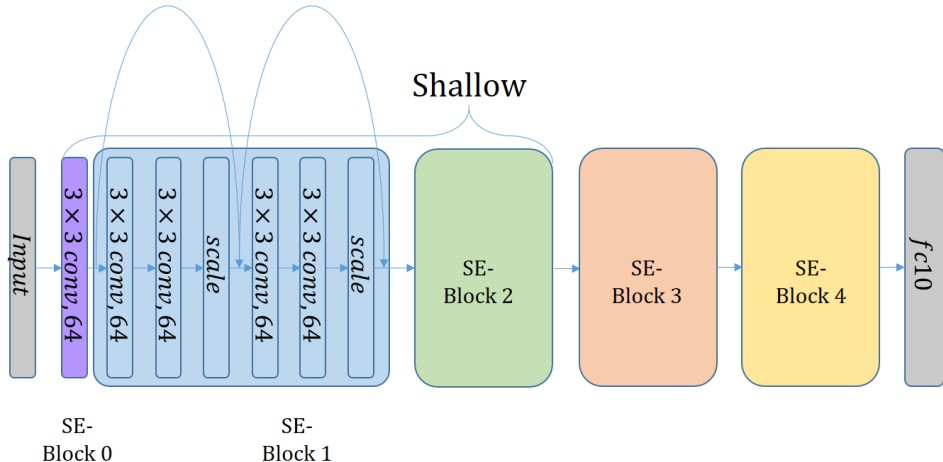

Figure 9: The architecture of SENet

## C.5 VGG-19

VGG-19 (Simonyan & Zisserman, 2014) is a typical neural network architecture with sixteen $3 \times 3$ convolutional layers and three fully-connected layers. We slightly modified the architecture by replacing the final 3 fully connected layers with 1 fully connected layer as is suggested by recent architectures. We apply Random Mask on the first four $3 \times 3$ convolutional layers.

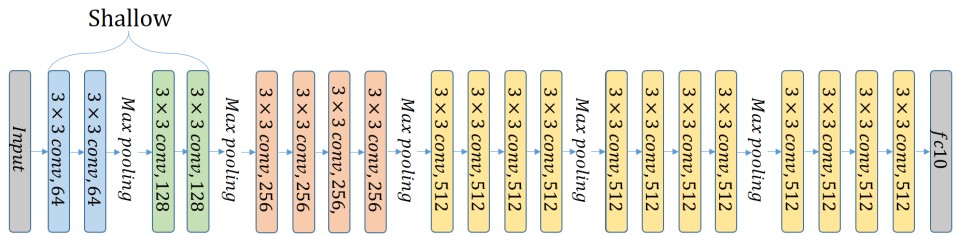

Figure 10: The architecture of VGG-19

# D  TRAINING PROCESS ON CIFAR-10 AND MNIST

To guarantee our experiments are reproducible, here we present more details on the training process in our experiments. When training models on CIFAR-10, we first subtract per-pixel mean. Then we apply a zero-padding of width $4$, a random horizontal flip and a random crop of size $32 \times 32$ on train data. No other data augmentation method is used. We apply SGD with momentum parameter $0.9$, weight decay parameter $5 \times 10^{-4}$ and mini-batch size $128$ to train on the data for $350$ epochs. The learning rate starts from $0.1$ and is divided by 10 when the number of epochs reaches $150$ and $250$. When training models on MNIST, we first subtract per-pixel mean. Then we apply random horizontal flip on train data. We apply SGD with momentum parameter $0.9$, weight decay parameter $5 \times 10^{-4}$ and mini-batch size $128$ to train on the data for $50$ epochs. The learning rate starts from $0.1$ and is divided by 10 when the number of epochs reaches $20$ and $40$. Figure 11 shows the train and test curves of a normal ResNet-18 and a Random Masked ResNet-18 on CIFAR-10 and MNIST. Different network structures share similar tendency in terms of the train and test curves.

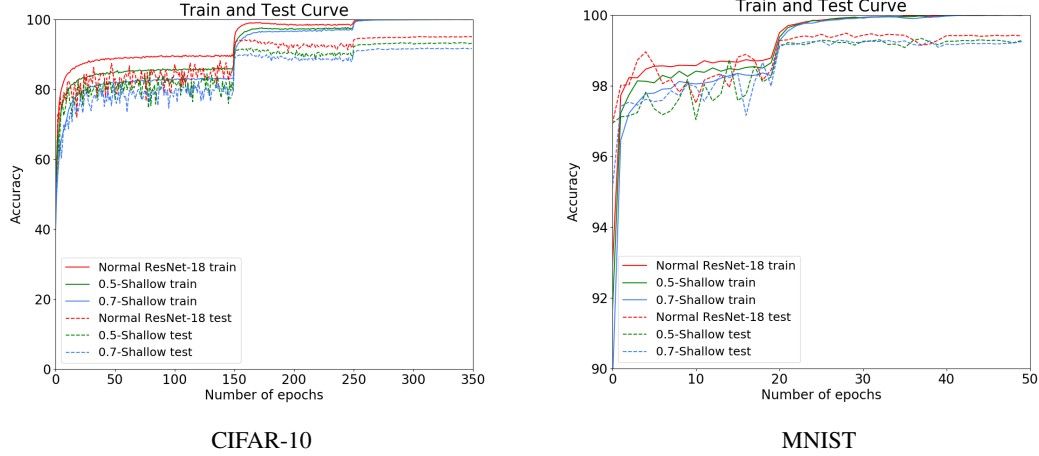

Figure 11: Train and test curve of normal ResNet-18 and Random Masked ResNet-18 on CIFAR-10 and MNIST.

# E    ADVERSARIAL EXAMPLES GENERATED BY APPLYING RANDOM MASK

## E.1    ADVERSARIAL EXAMPLES THAT CAN "FOOL" HUMAN

Figure 12 shows some adversarial examples generated from CIFAR-10 along with the corresponding original images. These examples are generated from CIFAR-10 against ResNet-18 with Random Mask of drop ratio 0.8 on the $0^{th}$, $1^{st}$, $2^{nd}$ blocks and another ResNet-18 with Random Mask of drop ratio 0.9 on the $1^{st}$, $2^{nd}$ blocks. We use attack method PGD with perturbation scale $\alpha = 16$ and $\alpha = 32$. We also show some adversarial examples generated from Tiny-ImageNet[1] along with the corresponding original images in Figure 13.

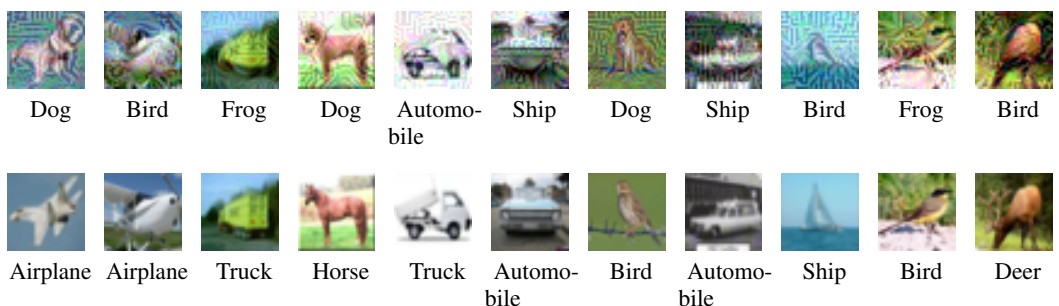

Figure 12: The adversarial examples (upper) shown in Figure 1 along with the original images (lower) from CIFAR-10.

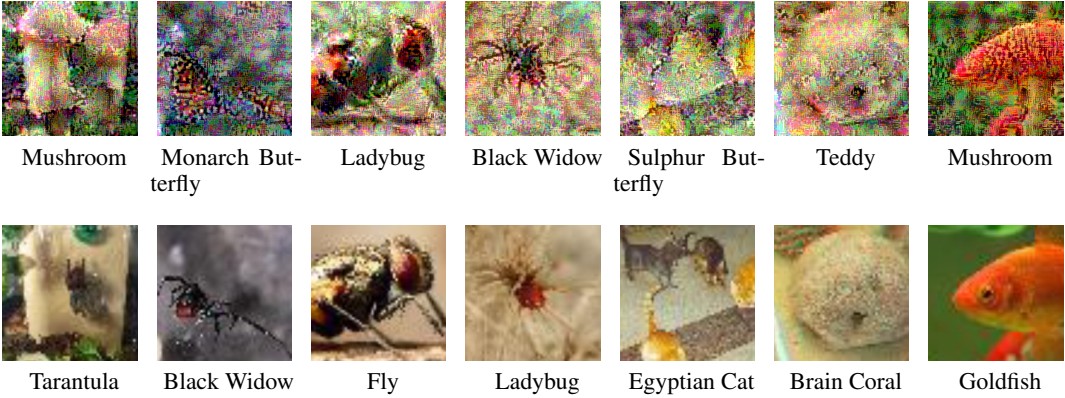

Figure 13: Adversarial examples (upper) generated from Tiny-ImageNet against ResNet-18 with Random Mask of ratio 0.9 on the $1^{st}$, $2^{nd}$ blocks. along with the original images (lower). The attack methods are PGD with scale 64 and 32, step size 1 and step number 40 and 80 respectively.

## E.2    RANDOMLY SELECTED ADVERSARIAL EXAMPLES

See Figure 15 for a randomly sampled set of images from Tiny-ImageNet along with the corresponding adversarial examples generated against ResNet-18 with Random Mask and normal ResNet-18.

---

[1] https://tiny-imagenet.herokuapp.com/

## F MORE EXPERIMENTAL RESULTS

### F.1 BLACK-BOX DEFENSE UNDER MADRY'S SETTING

Here we list the black-box settings in Madry's paper (Madry et al., 2017). In their experiments, ResNets are trained by minimizing the following loss:

$$\min_{\theta} \mathbb{E}_{(x,y)\sim\mathcal{D}} \left[ \max_{\delta\in\mathcal{S}} L(\theta, x + \delta, y) \right].$$

The outer minimization is achieved by gradient descent and the inner maximization is achieved by generating PGD adversarial examples with step size 2, the number of steps 7 and the perturbation scale 8. After training, in their black-box attack setting, they generate adversarial examples from naturally trained neural networks and test them on their models. Both FGSM and PGD adversarial examples have step size or perturbation scale 8 and PGD runs for 7 gradient descent steps with step size 2.

In Table 1, we apply Random Mask to shallow blocks with drop ratio 0.85. The ratio is selected by considering the trade-off of robustness and generalization performance, which is shown in Figure 14. When doing attacks, we generate the adversarial examples in the same way as Madry's paper (Madry et al., 2017) does.

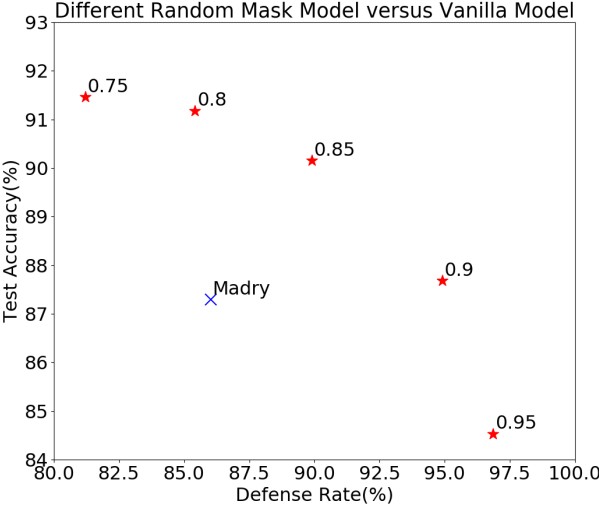

Figure 14: Relationship between defense rate against adversarial examples generated by PGD and test accuracy with respect to different drop ratios under Madry's setting (Madry et al., 2017). Each red star represents a specific drop ratio with its value written near the star. We can see the trade-off between robustness and generalization.

In this part, we apply Random Mask to five popular network structures - ResNet-18, ResNet-50, DenseNet-121, SENet-18, VGG-19, and test the black-box defense performance on CIFAR-10 and MNIST datasets.

Since both the intuition (see Section 2) and the extensive experiments (see Section 3.3 and Appendix F.5) show that we should apply Random Mask on the relatively shallow layers of the network structure, we would like to do so in this part of experiments. Illustrations of Random Mask applied to these network structures can be found in Appendix C. In addition, the detailed experiments on ResNet-18 (See Appendix F.5) show that defense performances are consistent against adversarial examples generated under different settings. Therefore, for brevity, we evaluate the defense performance on adversarial examples generated by PGD only in this subsection.

The results can be found in Table 4 and Table 5. Networks in the leftmost column are the target models which defend against adversarial examples. Networks in the first row are the source models to generate adversarial examples by PGD. 0.5-shallow and 0.7-shallow mean applying Random Mask with drop ratio 0.5 and 0.7 to the shallow layers of the network structure whose name lies just above them. The source and target networks are initialized differently if they share the same architecture. All the numbers except the Acc column mean the success rate of defense. The numbers in the Acc column mean the classification accuracy of the target model on clean test data. These results show that Random Mask can consistently improve the black-box defense performance of different network structures.

| Network Structure | ResNet-18 | ResNet-50 | DenseNet-121 | SENet-18 | VGG-19 | Acc |
|---|---|---|---|---|---|---|
| Normal ResNet-18 | 1.42% | 0.42% | 2.96% | 1.38% | 9.88% | 95.33% |
| 0.5-Shallow | 18.46% | 12.36% | 26.50% | 21.42% | 22.27% | 93.39% |
| 0.7-Shallow | 39.61% | 33.73% | 47.76% | 44.95% | 39.66% | 91.83% |
| Normal ResNet-50 | 4.33% | 0.10% | 3.95% | 3.93% | 17.58% | 95.25% |
| 0.5-Shallow | 16.79% | 5.24% | 20.48% | 19.70% | 24.82% | 94.43% |
| 0.7-Shallow | 32.69% | 19.56% | 38.34% | 37.58% | 37.99% | 93.46% |
| Normal DenseNet-121 | 4.10% | 0.58% | 0.60% | 3.24% | 10.11% | 95.53% |
| 0.5-Shallow | 12.67% | 4.18% | 10.89% | 13.30% | 19.22% | 93.97% |
| 0.7-Shallow | 23.79% | 14.54% | 25.44% | 27.90% | 29.61% | 92.82% |
| Normal SENet-18 | 1.10% | 0.52% | 2.29% | 0.62% | 8.02% | 95.09% |
| 0.5-Shallow | 17.47% | 11.39% | 23.69% | 19.62% | 22.66% | 93.53% |
| 0.7-Shallow | 34.19% | 27.27% | 43.13% | 39.05% | 34.92% | 92.54% |
| Normal VGG-19 | 6.94% | 4.23% | 9.98% | 6.76% | 4.63% | 93.93% |
| 0.5-Shallow | 29.73% | 25.86% | 38.66% | 35.80% | 26.93% | 91.73% |
| 0.7-Shallow | 49.94% | 45.74% | 57.74% | 56.17% | 46.10% | 90.11% |

Table 4: Black-box experiments on CIFAR-10. Networks in the leftmost column are the target models which defend against adversarial examples. Networks in the first row are the source models to generate adversarial examples by PGD. PGD runs for 20 steps with step size 1 and perturbation scale 16. 0.5-shallow and 0.7-shallow mean applying Random Mask with drop ratio 0.5 and 0.7 to the shallow layers of the network structure whose name lies just above them. All the numbers except the Acc column mean the success rate of defense.

| Network Structure | ResNet-18 | ResNet-50 | DenseNet-121 | SENet-18 | VGG-19 | Acc |
|---|---|---|---|---|---|---|
| Normal ResNet-18 | 0.06% | 13.80% | 2.34% | 0.10% | 8.69% | 99.49% |
| 0.5-Shallow | 3.45% | 17.18% | 7.41% | 4.49% | 17.01% | 99.34% |
| 0.7-Shallow | 15.28% | 47.68% | 19.29% | 21.05% | 38.54% | 99.29% |
| Normal ResNet-50 | 3.99% | 4.91% | 5.79% | 3.09% | 11.00% | 99.39% |
| 0.5-Shallow | 9.78% | 11.93% | 9.90% | 9.09% | 14.11% | 99.32% |
| 0.7-Shallow | 9.90% | 17.13% | 10.47% | 9.27% | 19.03% | 99.28% |
| Normal ResNet-50 | 1.24% | 18.10% | 0.04% | 1.83% | 10.36% | 99.48% |
| 0.5-Shallow | 3.09% | 18.34% | 1.52% | 3.99% | 13.19% | 99.46% |
| 0.7-Shallow | 3.59% | 36.63% | 2.18% | 5.78% | 20.76% | 99.38% |
| Normal SENet-18 | 0.52% | 14.31% | 3.17% | 0.08% | 12.51% | 99.41% |
| 0.5-Shallow | 3.33% | 16.85% | 5.83% | 1.73% | 13.67% | 99.35% |
| 0.7-Shallow | 8.84% | 26.97% | 9.36% | 8.97% | 18.96% | 99.32% |
| Normal VGG-19 | 4.09% | 33.19% | 6.25% | 6.36% | 2.45% | 99.48% |
| 0.5-Shallow | 9.54% | 33.97% | 9.12% | 13.90% | 9.56% | 99.37% |
| 0.7-Shallow | 21.00% | 37.68% | 20.92% | 26.44% | 30.10% | 99.34% |

Table 5: Black-box experiments on MNIST. Networks in the leftmost column are the target models which defend against adversarial examples. Networks in the first row are the source models to generate adversarial examples by PGD. PGD runs for 40 steps with step size $0.01 \times 255$ and perturbation scale $0.3 \times 255$. 0.5-shallow and 0.7-shallow mean applying Random Mask with drop ratio 0.5 and 0.7 to the shallow layers of the network structure whose name lies just above them. All the numbers except the Acc column mean the success rate of defense.

## F.3 WHITE-BOX

See Table 6 for the defense performance of ResNet-18 with Random Mask against white-box attacks on CIFAR-10 dataset. All the numbers except the Acc column mean the success rate of defense. The results on other network architectures are similar.

| Network Structure | $FGSM_1$ | $FGSM_2$ | $FGSM_4$ | $PGD_2$ | $PGD_4$ | $PGD_8$ | Acc |
|---|---|---|---|---|---|---|---|
| Normal ResNet-18 | 81.24% | 65.78% | 51.24% | 24.26% | 3.40% | 0.02% | 95.33% |
| 0.5-Shallow | 85.22% | 68.65% | 52.04% | 42.35% | 9.11% | 0.42% | 93.39% |
| 0.7-Shallow | 85.70% | 69.69% | 54.51% | 49.30% | 19.88% | 3.28% | 91.83% |

Table 6: White-box defense performance. $FGSM_1, FGSM_2, FGSM_4$ refer to FGSM with step size 1,2,4 respectively. $PGD_2, PGD_4, PGD_8$ refer to PGD with perturbation scale 2,4,8 and step number 4,6,10 respectively. The step size of all PGD are set to 1.

## F.4 TRANSFERABILITY AND GRAY-BOX DEFENSE

Here we show the gray-box defense ability of Random Mask and the transferability of the adversarial examples generated against Random Mask on CIFAR-10 dataset. We generate gray-box attacks in the following two ways. One way is to generate adversarial examples against one trained neural network and test those images on a network with the same structure but different initialization. The other way is specific to our Random Mask models. We generate adversarial examples on one trained network with Random Mask and test them on a network with the same drop ratio but different Random Mask. In both of these two ways, the adversarial

knows some information on the structure of the network, but does not know the parameters of it. To see the transferability of the generated adversarial examples, we also test them on DenseNet-121 and VGG-19.

| Source / Target | Normal ResNet-18 | 0.5-Shallow | 0.7-Shallow |
|---|---|---|---|
| Normal ResNet-18 | 13.91% | 20.90% | 28.42% |
| 0.5-Shallow | 28.83% | 20.22% | 23.24% |
| 0.5-Shallow$_{\text{DIF}}$ | 28.91% | 19.37% | 22.11% |
| 0.7-Shallow | 49.14% | 31.30% | 23.26% |
| 0.7-Shallow$_{\text{DIF}}$ | 48.23% | 31.77% | 23.54% |
| Normal DenseNet-121 | 20.23% | 22.38% | 28.05% |
| Normal VGG-19 | 15.83% | 17.43% | 20.85% |

Table 7: Results on gray-box attacks and transferability. We use FGSM with step size 16 to generate the adversarial examples on source networks and test them on target networks. For target networks, Normal ResNet-18, 0.5-Shallow and 0.7-Shallow represent the networks with the same structure as the corresponding source networks but with different initialization values. 0.5-Shallow$_{\text{DIF}}$ and 0.7-Shallow$_{\text{DIF}}$ represent the networks with the same drop ratios as the corresponding source networks but with different random masks.

Table 7 shows that Random Mask can also improve the performance under gray-box attacks. In addition, we find that CNNs with Random Mask have similar performance on adversarial examples generated by our two kinds of gray-box attacks. This phenomenon indicates that CNNs with Random Mask of same ratios have similar properties and catch similar information.

## F.5    FULL INFORMATION ON EXPERIMENTS MENTIONED IN SECTION 3.3

In this part, we will show more experimental results on Random Mask using different adversarial examples, different attack methods and different mask settings on ResNet-18. More specifically, we choose 5000 test images from CIFAR-10 which are correctly classified by the original network to generate FGSM and PGD adversarial examples, and 1000 test images for CW attack.

For FGSM, we try step size $\epsilon \in \{8, 16, 32\}$, namely **FGSM**$_8$, **FGSM**$_{16}$, **FGSM**$_{32}$, to generate adversarial examples. For PGD, we have tried more extensive settings. Let $\{\epsilon, T, \alpha\}$ be the PGD setting with step size $\epsilon$, the number of steps $T$ and the perturbation scale $\alpha$, then we have tried PGD settings $(1, 8, 4), (2, 4, 4), (4, 2, 4), (1, 12, 8), (2, 6, 8), (4, 3, 8), (1, 20, 16), (2, 10, 16), (4, 5, 16), (1, 40, 32), (2, 20, 32), (4, 10, 32)$ to generate PGD adversarial examples. From the experimental results, we observe the following phenomena. First, we find that the larger the perturbation scale is, the stronger the adversarial examples are. Second, for a fixed perturbation scale, the smaller the step size is, the more successful the attack is, as it searches the adversarial examples in a more careful way around the original image. Based on these observation, we only show strong PGD attack results in the Appendix, namely the settings $(1, 20, 16)$ (**PGD**$_{16}$), $(2, 10, 16)$ (**PGD**$_{2,16}$) and $(1, 40, 32)$ (**PGD**$_{32}$). Nonetheless, our models also perform much better on weak PGD attacks. For CW attack, we have also tried different confidence parameters $\kappa$. However, we find that for large $\kappa$, the algorithm is hard to find adversarial examples for some neural networks such as VGG because of its logit scale. For smaller $\kappa$, the adversarial examples have weak transfer ability, which means they can be easily defended even by normal networks. Therefore, in order to balance these two factors, we choose $\kappa = 40$ (**CW**$_{40}$) for DenseNet-121, ResNet-50, SENet-18 and $\kappa = 20$ (**CW**$_{20}$) for ResNet-18 as a good choice to compare our models with normal ones. The step number for choosing the parameter $c$ is set to 30.

Note that the noise of FGSM and PGD is considered in the sense of $\ell_\infty$ norm and the noise of CW is considered in the sense of $\ell_2$ norm. All adversarial examples used to evaluate can fool the original network. Table 8,9,10,11 and 12 list our experimental results. DC means we replace Random Mask with a decreased number of channels in the corresponding blocks to achieve the same drop ratio. SM means we use the same mask on all the channels in a layer. $\times n$ means we multiply the number of the channels in the corresponding blocks by $n$ times. EN means we ensemble five models with different masks of the same drop ratio.

| Network | $\textbf{FGSM}_8$ | $\textbf{FGSM}_{16}$ | $\textbf{FGSM}_{32}$ | $\textbf{PGD}_{16}$ | $\textbf{PGD}_{2,16}$ | $\textbf{PGD}_{32}$ | $\textbf{CW}_{40}$ | Acc |
|---|---|---|---|---|---|---|---|---|
| Normal ResNet-18 | 29.78% | 14.91% | 11.53% | 2.96% | 3.44% | 2.26% | 8.23% | 95.33% |
| 0.3-Shallow | 55.40% | 23.29% | 7.73% | 14.53% | 16.00% | 5.73% | 36.95% | 94.03% |
| 0.5-Shallow | 66.87% | 30.86% | 6.65% | 26.50% | 28.65% | 10.33% | 54.02% | 93.39% |
| 0.7-Shallow | 79.50% | 48.57% | 10.51% | 47.76% | 49.62% | 21.39% | 73.70% | 91.83% |
| 0.75-Shallow | 83.12% | 59.22% | 17.16% | 56.08% | 58.81% | 26.87% | 77.82% | 91.46% |
| 0.8-Shallow | 85.49% | 63.01% | 15.57% | 63.16% | 65.16% | 32.86% | 81.75% | 91.18% |
| 0.85-Shallow | 88.18% | 65.27% | 18.33% | 69.40% | 71.21% | 36.12% | 85.46% | 90.15% |
| 0.9-Shallow | 94.08% | 79.93% | 43.70% | 83.08% | 83.72% | 55.02% | 89.67% | 87.68% |
| 0.95-Shallow | 96.16% | 87.36% | 59.05% | 89.98% | 90.13% | 68.25% | 90.24% | 84.53% |
| 0.3-Deep | 28.51% | 14.62% | 8.78% | 1.95% | 2.43% | 1.30% | 7.88% | 95.16% |
| 0.5-Deep | 25.01% | 10.76% | 10.24% | 2.57% | 3.81% | 4.52% | 7.19% | 94.94% |
| 0.7-Deep | 23.94% | 11.23% | 10.48% | 3.24% | 4.07% | 2.64% | 10.10% | 94.61% |
| 0.5-Shallow, 0.5-Deep | 58.49% | 26.34% | 11.08% | 19.36% | 20.77% | 9.07% | 43.59% | 92.39% |
| 0.3-Shallow, 0.3-Deep | 51.03% | 24.15% | 10.82% | 12.67% | 14.06% | 6.75% | 29.65% | 94.16% |
| 0.3-Shallow, 0.7-Deep | 36.16% | 11.26% | 9.16% | 7.94% | 8.00% | 5.77% | 23.31% | 93.44% |
| 0.7-Shallow, 0.7-Deep | 64.85% | 27.43% | 10.09% | 32.72% | 33.84% | 16.26% | 62.47% | 89.78% |
| 0.7-Shallow, 0.3-Deep | 74.73% | 40.58% | 9.12% | 42.95% | 45.32% | 19.00% | 68.58% | 91.23% |
| 0.5-Shallow$_{\text{DC}}$ | 36.39% | 12.15% | 8.24% | 4.68% | 5.52% | 4.05% | 12.72% | 94.97% |
| 0.7-Shallow$_{\text{DC}}$ | 43.81% | 17.74% | 8.32% | 7.51% | 9.20% | 4.52% | 19.34% | 94.23% |
| 0.9-Shallow$_{\text{DC}}$ | 49.53% | 19.00% | 7.23% | 19.33% | 20.88% | 10.08% | 44.80% | 93.27% |
| 0.5-Shallow$_{\text{SM}}$ | 77.30% | 48.86% | 12.50% | 44.04% | 46.58% | 19.81% | 72.07% | 92.57% |
| 0.7-Shallow$_{\text{SM}}$ | 82.59% | 48.03% | 12.30% | 57.62% | 57.83% | 24.81% | 79.55% | 89.81% |
| 0.9-Shallow$_{\text{SM}}$ | 67.06% | 39.40% | 16.25% | 50.40% | 50.20% | 29.23% | 65.38% | 74.28% |
| 0.5-Shallow$_{\times 2}$ | 51.25% | 20.78% | 10.29% | 12.51% | 13.89% | 5.38% | 34.00% | 94.12% |
| 0.7-Shallow$_{\times 2}$ | 68.82% | 30.94% | 7.22% | 29.83% | 31.74% | 11.44% | 60.17% | 93.01% |
| 0.9-Shallow$_{\times 2}$ | 88.00% | 68.83% | 28.55% | 66.86% | 69.42% | 37.51% | 82.74% | 90.49% |
| 0.9-Shallow$_{\times 4}$ | 82.96% | 59.64% | 19.44% | 59.15% | 60.72% | 32.29% | 78.88% | 90.57% |
| Normal ResNet-18$_{\text{EN}}$ | 35.89% | 16.24% | 9.92% | 2.22% | 2.56% | 1.46% | 8.58% | 96.12% |
| 0.5-Shallow$_{\times 2,\text{EN}}$ | 55.25% | 19.84% | 8.36% | 11.86% | 13.44% | 5.30% | 37.37% | 95.24% |
| 0.5-Shallow$_{\text{EN}}$ | 69.35% | 31.38% | 7.73% | 27.58% | 29.68% | 9.97% | 58.07% | 94.56% |
| 0.7-Shallow$_{\text{EN}}$ | 81.98% | 51.81% | 8.57% | 50.79% | 53.88% | 23.28% | 77.74% | 93.31% |
| 0.85-Shallow$_{\text{EN}}$ | 90.13% | 66.51% | 17.91% | 71.38% | 72.30% | 38.25% | 87.37% | 91.77% |
| 0.9-Shallow$_{\text{EN}}$ | 95.37% | 81.95% | 43.42% | 85.14% | 85.79% | 56.02% | 91.36% | 89.45% |

Table 8: Extended experimental results of Section 3.3. Adversarial examples generated against *DenseNet-121*. The model trained on CIFAR-10 achieves 95.62% accuracy on test set. $\sigma$-Shallow$_{\text{DC}}$, $\sigma$-Shallow$_{\text{SM}}$, $\sigma$-Shallow$_{\times n}$ and $\sigma$-Shallow$_{\text{EN}}$ mean dropping channels with ratio $\sigma$, applying same mask with ratio $\sigma$, increasing channel number to $n$ times with mask ratio $\sigma$ for every channel and ensemble five models with different masks of same ratio $\sigma$ respectively. The entries in the middle seven columns are success rates of defense under different settings.

| Network | FGSM$_8$ | FGSM$_{16}$ | FGSM$_{32}$ | PGD$_{16}$ | PGD$_{2,16}$ | PGD$_{32}$ | CW$_{20}$ | Acc |
|---|---|---|---|---|---|---|---|---|
| Normal ResNet-18 | 26.99% | 13.91% | 3.57% | 1.42% | 1.84% | 0.96% | 2.19% | 95.33% |
| 0.3-Shallow | 48.76% | 21.32% | 9.54% | 8.14% | 9.51% | 4.02% | 38.87% | 94.03% |
| 0.5-Shallow | 59.66% | 30.48% | 11.60% | 18.46% | 21.44% | 7.70% | 60.65% | 93.39% |
| 0.7-Shallow | 74.00% | 47.11% | 15.65% | 39.61% | 43.17% | 16.09% | 79.04% | 91.83% |
| 0.75-Shallow | 78.37% | 56.05% | 21.44% | 49.14% | 52.21% | 20.31% | 81.59% | 91.46% |
| 0.8-Shallow | 81.67% | 59.14% | 19.60% | 55.84% | 59.63% | 26.61% | 82.78% | 91.18% |
| 0.85-Shallow | 86.31% | 63.16% | 22.23% | 64.73% | 67.03% | 31.33% | 86.06% | 90.15% |
| 0.9-Shallow | 92.89% | 77.90% | 45.63% | 81.70% | 82.50% | 54.12% | 90.29% | 87.68% |
| 0.95-Shallow | 95.07% | 85.40% | 59.91% | 88.31% | 89.64% | 66.52% | 90.97% | 84.53% |
| 0.3-Deep | 25.96% | 15.46% | 7.18% | 1.18% | 1.28% | 0.88% | 2.66% | 95.16% |
| 0.5-Deep | 25.21% | 9.21% | 1.44% | 2.17% | 2.63% | 2.31% | 3.15% | 94.94% |
| 0.7-Deep | 24.36% | 9.49% | 2.60% | 2.36% | 3.08% | 1.31% | 6.62% | 94.61% |
| 0.5-Shallow, 0.5-Deep | 53.46% | 24.65% | 7.08% | 13.48% | 15.65% | 6.99% | 49.95% | 92.39% |
| 0.3-Shallow, 0.3-Deep | 43.32% | 20.55% | 4.14% | 7.31% | 9.36% | 4.46% | 32.92% | 94.16% |
| 0.3-Shallow, 0.7-Deep | 34.09% | 11.05% | 1.58% | 6.01% | 6.77% | 4.56% | 24.11% | 93.44% |
| 0.7-Shallow, 0.7-Deep | 61.22% | 28.11% | 13.78% | 27.12% | 30.24% | 13.51% | 69.15% | 89.78% |
| 0.7-Shallow, 0.3-Deep | 70.43% | 39.15% | 13.94% | 36.88% | 39.81% | 15.45% | 74.57% | 91.23% |
| 0.5-Shallow$_{DC}$ | 32.86% | 13.89% | 3.71% | 1.93% | 2.89% | 2.19% | 6.10% | 94.97% |
| 0.7-Shallow$_{DC}$ | 37.96% | 16.23% | 5.05% | 4.30% | 5.96% | 2.65% | 15.44% | 94.23% |
| 0.9-Shallow$_{DC}$ | 48.54% | 19.10% | 11.37% | 14.34% | 16.01% | 7.04% | 50.62% | 93.27% |
| 0.5-Shallow$_{SM}$ | 73.96% | 47.63% | 16.60% | 36.19% | 40.86% | 15.52% | 73.68% | 92.57% |
| 0.7-Shallow$_{SM}$ | 80.80% | 48.37% | 15.26% | 53.69% | 54.78% | 22.90% | 82.34% | 89.81% |
| 0.9-Shallow$_{SM}$ | 69.15% | 43.55% | 20.26% | 50.68% | 50.80% | 28.82% | 71.62% | 74.28% |
| 0.5-Shallow$_{\times 2}$ | 46.50% | 21.37% | 6.06% | 6.86% | 8.65% | 3.59% | 39.12% | 94.12% |
| 0.7-Shallow$_{\times 2}$ | 63.37% | 29.90% | 12.07% | 20.70% | 24.08% | 7.76% | 67.02% | 93.01% |
| 0.9-Shallow$_{\times 2}$ | 84.28% | 64.47% | 31.90% | 62.65% | 65.24% | 33.06% | 85.08% | 90.49% |
| 0.9-Shallow$_{\times 4}$ | 78.24% | 56.31% | 23.28% | 52.38% | 55.63% | 25.91% | 82.26% | 90.57% |
| Normal ResNet-18$_{EN}$ | 29.66% | 14.37% | 3.97% | 0.98% | 1.24% | 0.54% | 2.00% | 96.12% |
| 0.5-Shallow$_{\times 2,EN}$ | 49.16% | 19.81% | 6.73% | 7.02% | 8.50% | 3.46% | 38.12% | 95.24 % |
| 0.5-Shallow$_{EN}$ | 63.38% | 30.25% | 11.05% | 18.15% | 21.14% | 6.71% | 63.90% | 94.56% |
| 0.7-Shallow$_{EN}$ | 77.25% | 50.07% | 13.80% | 41.59% | 44.78% | 15.85% | 80.86% | 93.31% |
| 0.85-Shallow$_{EN}$ | 88.56% | 65.23% | 22.50% | 65.68% | 68.31% | 32.77% | 88.26% | 91.77% |
| 0.9-Shallow$_{EN}$ | 94.31% | 79.47% | 44.67% | 82.97% | 84.05% | 54.46% | 90.52% | 89.4 % |

Table 9: Extended experimental results of Section 3.3. Adversarial examples are generated against *ResNet-18*. The model trained on CIFAR-10 achieves 95.27% accuracy on test set. $\sigma$-Shallow$_{DC}$, $\sigma$-Shallow$_{SM}$, $\sigma$-Shallow$_{\times n}$ and $\sigma$-Shallow$_{EN}$ mean dropping channels with ratio $\sigma$, applying same mask with ratio $\sigma$, increasing channel number to $n$ times with mask ratio $\sigma$ for every channel and ensemble five models with different masks of same ratio $\sigma$ respectively. The entries in the middle seven columns are success rates of defense under different settings.

| Network | FGSM$_8$ | FGSM$_{16}$ | FGSM$_{32}$ | PGD$_{16}$ | PGD$_{2,16}$ | PGD$_{32}$ | CW$_{40}$ | Acc |
|---|---|---|---|---|---|---|---|---|
| Normal ResNet-18 | 29.33% | 15.14% | 3.88% | 0.42% | 0.96% | 0.08% | 0.00% | 95.33% |
| 0.3-Shallow | 45.32% | 18.89% | 9.16% | 4.36% | 5.81% | 0.92% | 1.98% | 94.03% |
| 0.5-Shallow | 56.26% | 27.32% | 10.72% | 12.36% | 15.29% | 3.48% | 8.92% | 93.39% |
| 0.7-Shallow | 70.57% | 42.40% | 14.98% | 33.73% | 37.56% | 12.38% | 33.08% | 91.83% |
| 0.75-Shallow | 77.18% | 53.01% | 19.68% | 42.67% | 46.72% | 15.88% | 39.10% | 91.46% |
| 0.8-Shallow | 80.33% | 56.21% | 18.03% | 52.45% | 55.54% | 22.17% | 47.52% | 91.18% |
| 0.85-Shallow | 84.81% | 61.02% | 21.50% | 62.62% | 63.80% | 29.35% | 53.71% | 90.15% |
| 0.9-Shallow | 92.17% | 77.68% | 45.93% | 81.20% | 82.50% | 53.44% | 66.70% | 87.68% |
| 0.95-Shallow | 94.43% | 85.54% | 60.71% | 88.63% | 89.16% | 66.69% | 71.82% | 84.53% |
| 0.3-Deep | 27.78% | 15.03% | 8.07% | 0.42% | 0.60% | 0.12% | 0.00% | 95.16% |
| 0.5-Deep | 27.24% | 10.29% | 2.47% | 0.52% | 1.10% | 0.50% | 0.00% | 94.94% |
| 0.7-Deep | 24.81% | 9.99% | 2.50% | 0.67% | 1.01% | 0.20% | 0.00% | 94.61% |
| 0.5-Shallow, 0.5-Deep | 48.78% | 21.04% | 6.66% | 7.51% | 9.79% | 2.52% | 5.09% | 92.39% |
| 0.3-Shallow, 0.3-Deep | 42.18% | 18.20% | 5.26% | 3.96% | 5.44% | 1.65% | 1.64% | 94.16% |
| 0.3-Shallow, 0.7-Deep | 33.11% | 11.08% | 2.27% | 2.45% | 3.54% | 1.29% | 0.55% | 93.44% |
| 0.7-Shallow, 0.7-Deep | 56.39% | 24.14% | 12.18% | 21.86% | 24.88% | 9.01% | 22.25% | 89.78% |
| 0.7-Shallow, 0.3-Deep | 66.33% | 36.31% | 13.09% | 30.13% | 33.96% | 12.09% | 30.68% | 91.23% |
| 0.5-Shallow$_{DC}$ | 31.56% | 13.64% | 4.87% | 0.80% | 1.30% | 0.28% | 0.11% | 94.97% |
| 0.7-Shallow$_{DC}$ | 37.52% | 15.72% | 5.38% | 1.79% | 2.91% | 0.56% | 0.44% | 94.23% |
| 0.9-Shallow$_{DC}$ | 44.00% | 16.90% | 10.30% | 8.93% | 11.35% | 3.46% | 4.95% | 93.27% |
| 0.5-Shallow$_{SM}$ | 69.40% | 41.82% | 14.27% | 29.83% | 33.51% | 9.60% | 26.65% | 92.57% |
| 0.7-Shallow$_{SM}$ | 77.25% | 44.94% | 13.80% | 49.52% | 51.13% | 21.24% | 46.44% | 89.81% |
| 0.9-Shallow$_{SM}$ | 64.32% | 39.76% | 19.21% | 50.16% | 49.15% | 28.24% | 45.18% | 74.28% |
| 0.5-Shallow$_{\times2}$ | 41.51% | 18.47% | 6.02% | 3.67% | 4.80% | 0.62% | 1.32% | 94.12% |
| 0.7-Shallow$_{\times2}$ | 58.59% | 25.92% | 11.20% | 14.75% | 18.29% | 4.34% | 13.77% | 93.01% |
| 0.9-Shallow$_{\times2}$ | 83.05% | 63.73% | 29.22% | 58.85% | 61.82% | 28.09% | 50.11% | 90.49% |
| 0.9-Shallow$_{\times4}$ | 75.74% | 55.03% | 21.59% | 48.93% | 51.78% | 20.52% | 47.08% | 90.57% |
| Normal ResNet-18$_{EN}$ | 32.70% | 15.49% | 4.93% | 0.32% | 0.84% | 0.06% | 0.00% | 96.12% |
| 0.5-Shallow$_{\times2,EN}$ | 44.90% | 16.55% | 6.41% | 2.70% | 4.00% | 0.64% | 1.42% | 95.24% |
| 0.5-Shallow$_{EN}$ | 59.64% | 26.21% | 10.17% | 11.41% | 14.23% | 2.48% | 8.12% | 94.56% |
| 0.7-Shallow$_{EN}$ | 73.45% | 45.60% | 12.99% | 33.62% | 38.49% | 12.06% | 32.60% | 93.31% |
| 0.85-Shallow$_{EN}$ | 87.58% | 62.24% | 21.81% | 62.84% | 64.88% | 29.10% | 54.29% | 91.77% |
| 0.9-shallow$_{en}$ | 93.87% | 79.15% | 46.44% | 82.71% | 83.32% | 53.87% | 67.71% | 89.45% |

Table 10: Extended experimental results of Section 3.3. Adversarial examples are generated against *ResNet-50*. The model trained on CIFAR-10 achieves 95.69% accuracy on test set. $\sigma$-Shallow$_{DC}$, $\sigma$-Shallow$_{SM}$, $\sigma$-Shallow$_{\times n}$ and $\sigma$-Shallow$_{EN}$ mean dropping channels with ratio $\sigma$, applying same mask with ratio $\sigma$, increasing channel number to $n$ times with mask ratio $\sigma$ for every channel and ensemble five models with different masks of same ratio $\sigma$ respectively. The entries in the middle seven columns are success rates of defense under different settings.

| Network | FGSM$_8$ | FGSM$_{16}$ | FGSM$_{32}$ | PGD$_{16}$ | PGD$_{2,16}$ | PGD$_{32}$ | CW$_{40}$ | Acc |
|---|---|---|---|---|---|---|---|---|
| Normal ResNet-18 | 25.53% | 17.47% | 8.56% | 1.38% | 1.78% | 0.84% | 0.00% | 95.33% |
| 0.3-Shallow | 46.12% | 23.30% | 10.48% | 9.57% | 10.19% | 4.44% | 2.66% | 94.03% |
| 0.5-Shallow | 57.05% | 31.01% | 11.07% | 21.42% | 23.17% | 7.90% | 14.61% | 93.39% |
| 0.7-Shallow | 72.67% | 48.17% | 15.20% | 44.95% | 46.90% | 19.67% | 39.89% | 91.83% |
| 0.75-Shallow | 78.23% | 58.19% | 21.20% | 53.56% | 55.32% | 23.24% | 47.33% | 91.46% |
| 0.8-Shallow | 82.27% | 61.61% | 19.70% | 61.55% | 63.83% | 31.05% | 51.14% | 91.18% |
| 0.85-Shallow | 85.80% | 65.92% | 22.73% | 69.82% | 70.55% | 34.95% | 57.36% | 90.15% |
| 0.9-Shallow | 92.93% | 79.13% | 48.34% | 84.55% | 84.63% | 55.58% | 65.63% | 87.68% |
| 0.95-Shallow | 94.77% | 87.13% | 63.36% | 90.63% | 90.61% | 69.07% | 69.14% | 84.53% |
| 0.3-Deep | 23.76% | 16.66% | 9.52% | 1.12% | 1.26% | 0.66% | 0.00% | 95.16% |
| 0.5-Deep | 23.01% | 12.19% | 6.56% | 1.73% | 2.05% | 2.05% | 0.00% | 94.94% |
| 0.7-Deep | 22.87% | 11.61% | 6.63% | 2.12% | 2.54% | 1.29% | 0.19% | 94.61% |
| 0.5-Shallow, 0.5-Deep | 51.20% | 25.49% | 10.43% | 15.05% | 17.41% | 7.43% | 12.26% | 92.39% |
| 0.3-Shallow, 0.3-Deep | 42.34% | 22.07% | 7.84% | 8.47% | 9.68% | 4.44% | 1.70% | 94.16% |
| 0.3-Shallow, 0.7-Deep | 31.43% | 12.17% | 6.34% | 6.19% | 6.25% | 4.82% | 1.53% | 93.44% |
| 0.7-Shallow, 0.7-Deep | 57.26% | 29.36% | 13.99% | 31.68% | 32.58% | 15.16% | 30.00% | 89.78% |
| 0.7-Shallow, 0.3-Deep | 68.66% | 41.32% | 13.72% | 40.56% | 42.30% | 17.58% | 35.71% | 91.23% |
| 0.5-Shallow$_{DC}$ | 30.81% | 14.77% | 6.08% | 2.13% | 2.61% | 1.83% | 0.00% | 94.97% |
| 0.7-Shallow$_{DC}$ | 34.57% | 17.32% | 8.04% | 3.65% | 4.54% | 2.07% | 0.19% | 94.23% |
| 0.9-Shallow$_{DC}$ | 43.46% | 17.61% | 10.54% | 15.15% | 15.51% | 7.12% | 7.41% | 93.27% |
| 0.5-Shallow$_{SM}$ | 71.27% | 49.21% | 16.27% | 41.34% | 43.00% | 16.93% | 34.92% | 92.57% |
| 0.7-Shallow$_{SM}$ | 79.48% | 49.66% | 15.65% | 58.96% | 60.00% | 26.78% | 48.28% | 89.81% |
| 0.9-Shallow$_{SM}$ | 65.85% | 42.59% | 21.87% | 52.63% | 52.91% | 30.38% | 43.22% | 74.28% |
| 0.5-Shallow$_{\times 2}$ | 44.13% | 21.71% | 9.49% | 8.91% | 9.61% | 3.47% | 2.65% | 94.12% |
| 0.7-Shallow$_{\times 2}$ | 60.51% | 30.89% | 11.58% | 24.82% | 26.79% | 8.36% | 21.52% | 93.01% |
| 0.9-Shallow$_{\times 2}$ | 85.26% | 67.91% | 32.51% | 67.34% | 69.14% | 36.78% | 52.96% | 90.49% |
| 0.9-Shallow$_{\times 4}$ | 78.65% | 60.05% | 24.45% | 57.88% | 59.63% | 29.79% | 50.19% | 90.57% |
| Normal ResNet-18$_{EN}$ | 27.87% | 17.80% | 8.81% | 1.02% | 1.24% | 0.52% | 0.00% | 96.12% |
| 0.5-Shallow$_{\times 2,EN}$ | 45.04% | 20.25% | 9.69% | 7.46% | 9.02% | 3.30% | 2.84% | 95.24% |
| 0.5-Shallow$_{EN}$ | 60.42% | 31.08% | 10.77% | 20.98% | 22.56% | 7.09% | 13.61% | 94.56% |
| 0.7-Shallow$_{EN}$ | 76.08% | 51.49% | 13.19% | 46.98% | 49.31% | 18.95% | 39.51% | 93.31% |
| 0.85-Shallow$_{EN}$ | 88.64% | 67.26% | 23.30% | 71.44% | 72.42% | 37.16% | 56.36% | 91.77% |
| 0.9-Shallow$_{EN}$ | 94.40% | 81.32% | 48.52% | 85.81% | 86.28% | 56.14% | 66.99% | 89.45% |

Table 11: Extended experimental results of Section 3.3. Adversarial examples are generated against *SENet-18*. The model trained on CIFAR-10 achieves 95.15% accuracy on test set. $\sigma$-Shallow$_{DC}$, $\sigma$-Shallow$_{SM}$, $\sigma$-Shallow$_{\times n}$ and $\sigma$-Shallow$_{EN}$ mean dropping channels with ratio $\sigma$, applying same mask with ratio $\sigma$, increasing channel number to $n$ times with mask ratio $\sigma$ for every channel and ensemble five models with different masks of same ratio $\sigma$ respectively. The entries in the middle seven columns are success rates of defense under different settings.

| Network | FGSM$_8$ | FGSM$_{16}$ | FGSM$_{32}$ | PGD$_{16}$ | PGD$_{2,16}$ | PGD$_{32}$ | Acc |
|---|---|---|---|---|---|---|---|
| Normal ResNet-18 | 37.67% | 20.25% | 5.40% | 9.88% | 12.81% | 6.72% | 95.33% |
| 0.3-Shallow | 50.06% | 23.54% | 9.53% | 14.99% | 18.66% | 9.28% | 94.03% |
| 0.5-Shallow | 57.35% | 30.52% | 11.13% | 22.27% | 26.87% | 10.85% | 93.39% |
| 0.7-Shallow | 71.75% | 47.35% | 15.47% | 39.66% | 44.42% | 15.91% | 91.83% |
| 0.75-Shallow | 76.81% | 56.69% | 19.44% | 47.88% | 53.76% | 18.76% | 91.46% |
| 0.8-Shallow | 79.46% | 61.45% | 21.36% | 56.21% | 61.01% | 23.16% | 91.18% |
| 0.85-Shallow | 85.51% | 66.55% | 25.35% | 64.09% | 67.77% | 27.53% | 90.15% |
| 0.9-Shallow | 92.58% | 80.68% | 51.90% | 81.65% | 83.90% | 52.47% | 87.68% |
| 0.95-Shallow | 95.24% | 87.10% | 64.22% | 88.50% | 90.04% | 64.64% | 84.53% |
| 0.3-Deep | 36.72% | 18.97% | 9.65% | 9.21% | 11.48% | 6.40% | 95.16% |
| 0.5-Deep | 35.93% | 13.80% | 2.99% | 10.24% | 12.77% | 8.95% | 94.94% |
| 0.7-Deep | 34.05% | 13.06% | 4.04% | 10.36% | 12.70% | 7.46% | 94.61% |
| 0.5-Shallow, 0.5-Deep | 52.05% | 24.88% | 6.74% | 17.83% | 21.75% | 10.37% | 92.39% |
| 0.3-Shallow, 0.3-Deep | 47.36% | 22.74% | 5.04% | 14.44% | 17.82% | 9.08% | 94.16% |
| 0.3-Shallow, 0.7-Deep | 40.38% | 13.50% | 3.28% | 11.38% | 14.52% | 8.12% | 93.44% |
| 0.7-Shallow, 0.7-Deep | 59.19% | 28.00% | 12.13% | 29.88% | 34.28% | 14.12% | 89.78% |
| 0.7-Shallow, 0.3-Deep | 67.14% | 40.57% | 13.80% | 37.63% | 42.54% | 16.00% | 91.23% |
| 0.5-Shallow$_{DC}$ | 37.37% | 16.99% | 6.62% | 9.76% | 12.35% | 8.07% | 94.97% |
| 0.7-Shallow$_{DC}$ | 42.39% | 19.90% | 6.74% | 11.93% | 15.39% | 8.01% | 94.23% |
| 0.9-Shallow$_{DC}$ | 47.41% | 21.12% | 11.43% | 20.10% | 23.16% | 10.82% | 93.27% |
| 0.5-Shallow$_{SM}$ | 69.61% | 46.57% | 14.85% | 37.31% | 43.96% | 15.26% | 92.57% |
| 0.7-Shallow$_{SM}$ | 79.69% | 48.86% | 13.87% | 52.11% | 56.12% | 20.35% | 89.81% |
| 0.9-Shallow$_{SM}$ | 67.77% | 44.38% | 20.74% | 49.64% | 51.28% | 27.31% | 74.28% |
| 0.5-Shallow$_{\times 2}$ | 46.93% | 21.74% | 7.11% | 14.52% | 18.17% | 8.09% | 94.12% |
| 0.7-Shallow$_{\times 2}$ | 60.23% | 29.72% | 11.07% | 23.52% | 28.11% | 11.56% | 93.01% |
| 0.9-Shallow$_{\times 2}$ | 83.32% | 66.44% | 33.11% | 61.73% | 66.76% | 30.33% | 90.49% |
| 0.9-Shallow$_{\times 4}$ | 77.68% | 58.73% | 24.40% | 52.07% | 57.72% | 23.55% | 90.57% |
| Normal ResNet-18$_{EN}$ | 39.42% | 19.53% | 6.69% | 8.67% | 12.13% | 6.46% | 96.12% |
| 0.5-Shallow$_{\times 2,EN}$ | 49.51% | 19.29% | 7.16% | 13.71% | 17.73% | 8.74% | 95.24% |
| 0.5-Shallow$_{EN}$ | 60.43% | 31.07% | 10.50% | 21.54% | 27.09% | 10.99% | 94.56% |
| 0.7-Shallow$_{EN}$ | 74.11% | 50.89% | 13.54% | 41.26% | 47.68% | 16.01% | 93.31% |
| 0.85-Shallow$_{EN}$ | 87.48% | 67.75% | 25.62% | 65.68% | 69.68% | 29.26% | 91.77% |
| 0.9-Shallow$_{EN}$ | 94.14% | 82.46% | 52.59% | 83.05% | 85.46% | 53.42% | 89.45% |

Table 12: Extended experimental results of Section 3.3. Adversarial examples are generated against *VGG-19*. The model trained on CIFAR-10 achieves 94.04% accuracy on test set. $\sigma$-Shallow$_{DC}$, $\sigma$-Shallow$_{SM}$, $\sigma$-Shallow$_{\times n}$ and $\sigma$-Shallow$_{EN}$ mean dropping channels with ratio $\sigma$, applying same mask with ratio $\sigma$, increasing channel number to $n$ times with mask ratio $\sigma$ for every channel and ensemble five models with different masks of same ratio $\sigma$ respectively. The entries in the middle six columns are success rates of defense under different settings.

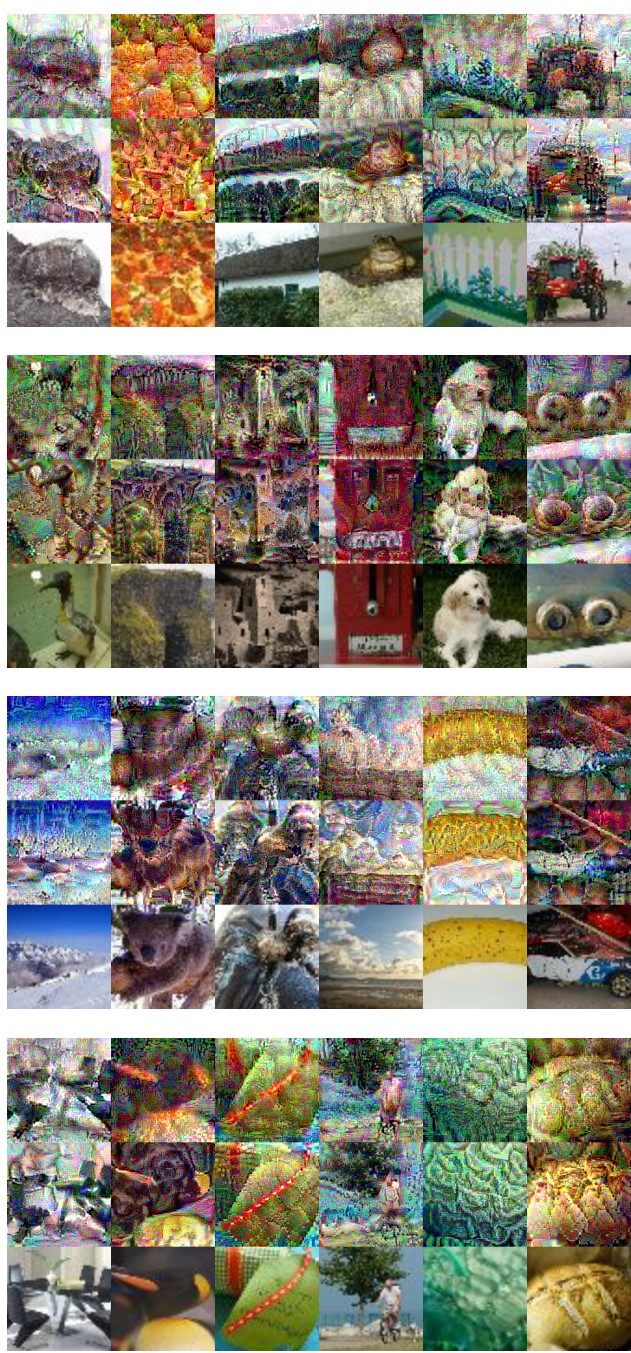

Figure 15: Randomly sampled images from Tiny-ImageNet dataset. The network structure used to generate these images is ResNet-18 with Random Mask of ratio $0.9$ on the $1^{st}$, $2^{nd}$ blocks. The attack method is PGD with perturbation scale $64$, step size $1$ and step number $80$. For each image, we show the image generated against network with Random Mask (upper), the image generated against the normal ResNet-18 (middle) and the original image (lower).

