# OpenReview forum: "RANDOM MASK: Towards Robust Convolutional Neural Networks"
_ICLR.cc/2019/Conference_

### Official Review · AnonReviewer3 · 2018-11-02
**Simple approach with experimental validations, however, seems ad hoc**

**Rating:** 4
**Confidence:** 3

**Review:**

I am upgrading my reviews after the rebuttal, which actually has convinced me that there is something interesting going on in this paper. However, I'm not entirely convinced as the approach seems to be ad hoc. the intuitions provided are somewhat satisfactory, but it's not clear why the method works.. for example, the approach is highly sensitive to the hyperparameter "drop rate" and there is no way to find a good value for it. I'm inclined towards rejection as, even though results are almost satisfying, I yet don't understand what exactly is happening. Most of the arguments seems to be handwavy. I personally feel like a paper as simple as this one with not enough conceptual justifications, but good results (like this one), should go to a workshop.

======
The authors propose to randomly drop a few parameters at the beginning and fix the resulting architecture for train and test. The claim is that the resulting network is robust to adversarial attacks.

Major concerns:
An extremely simple approach of pruning neural networks (randomly dropping weights) with no justification whatsoever. There are so many other network pruning papers available. If the point is to use pruned network then the authors must provide analysis over other pruning schemes as well.

Another major concern (technical contributions): How is the idea of randomly dropping weights different from Deep Expander Networks (Prabhu et al., ECCV 2018)? Please clarify.

Minor suggestion: Another simple approach to test the hypotheses would be to try dropout at test time and see the performance.

---

> ### Author Response · Authors · 2018-11-22
> **Random Mask does NOT reduce the number of parameters**
>
> It seems there are misunderstandings of our method. For your two major concerns, we first give a brief answer and then provide detailed explanations.
>
> 1. Random Mask is NOT a weight-dropping pruning method.
> 2. We conduct experiments comparing Random Mask with pruning methods (both you mentioned and those commonly used). It turns out a network with Random mask is far more robust than pruning.
>
>
> Detailed Explanations:
>
> 1. Random Mask is a simple but carefully designed method. It removes some nodes in the shallow convolutional layers whose receptive fields are relatively small. This is very different from typical pruning methods which drop weights, remove channels or restrict connections between channels in two adjacent layers. The key idea of Random Mask is that by removing a part of such neurons, the remaining neurons in the shallow layers can not only response to features, but also automatically record the locations of the features. As a result, these remaining neurons in shallow layers together detect the spatial structure of the features, much better than the standard neural networks.
>
> The motivation of our design comes from the recent observation of adversarial examples. In many cases, the adversarial examples change a patch of the original image so that the perturbed patch looks like a small part of the incorrectly classified object. This perturbed patch, although contains crucial features of the incorrectly classified object, usually appears at the wrong location and does not have the right spatial structure with other parts of the image. For example, the adversarial example of a panda image is misclassified as a monkey because a patch of the panda skin is perturbed adversarially so that it alone looks like the monkey’s face. However, this patch does not form a right structure of a monkey with other parts of the images (see Figure 11 in [1]).
>
> In sum, current deep neural networks are strong at detecting features, but relatively weak at telling if the spatial location/structure is right. Random Mask tries to strengthen the ability of neural networks in utilizing the spatial information.
>
> 2.The experimental results comparing Random Mask with typical pruning methods are given below. Common pruning methods do not improve the robustness of neural networks significantly, while a network with Random Mask is far more robust.
>
> We test the black-box defense ability of a ResNet-18 with an expander graph compressing all connections between channels by a factor of 2 (following the method proposed in [2]). For comparison, we also list the performance of a ResNet-18 which prunes whole channels in the shallow layers (i.e. Shallow_{DC} in our paper) and a ResNet-18 equipped with Random Mask. The results are listed in the table below. Networks in the first row are the source models to generate adversarial examples by PGD with perturbation scale of 16, step size of 1 and 20 steps.
>
> ----------------------------------------------------------------------------
> |	                            |  DenseNet  |   SENet    |   TestAcc |
> | Normal  ResNet    |      2.96%      |    1.38%   |   95.33% |
> | Expander ResNet |  	  3.13%       |    1.46%   |   94.99% |
> | Pruning Channel  |     4.68%       |    2.13%   |   94.97% |
> | Random Mask       |   26.50%      |   21.42%  |   93.39% |
> ----------------------------------------------------------------------------
>
> As for your suggestion, we are not sure what hypotheses you hoped to verify by trying dropout at test time. Nonetheless, we think trying dropping at test time is similar to Stochastic Activation Pruning ([3]). In their work, they tested SAP in terms of black-box defense (Figure 1 (c) SAP-100 vs Dense in [3]), yet the performance is not as good as our results when the perturbation scale is 8, 16 and 32. Also, directly dropping at test time and scaling up the remaining will incur a significant drop in test accuracy. We tried to mask out 50% of the neurons in the shallow blocks of a ResNet-18 at test time only, and scale up the rest by a factor of 2. It turned out that the test accuracy dropped to around 20%, which is not acceptable.
>
> Further discussion is welcomed if our reply does not address your concerns.
>
> [1]Liu, Mengchen, et al. "Analyzing the Noise Robustness of Deep Neural Networks." arXiv preprint arXiv:1810.03913(2018).
> [2]Prabhu, Ameya, Girish Varma, and Anoop Namboodiri. "Deep Expander Networks: Efficient Deep Networks from Graph Theory." arXiv preprint arXiv:1711.08757 (2017).
> [3]Dhillon, Guneet S., et al. "Stochastic activation pruning for robust adversarial defense." arXiv preprint arXiv:1803.01442 (2018).

---

> ### Author Response · Authors · 2018-12-06
> **Not ad hoc, Random Mask improves robustness of existing CNN architectures**
>
> Thanks for updating your review.
>
> > the approach is highly sensitive to the hyperparameter "drop rate" and there is no way to find a good value for it.
>
> Random Mask is not highly sensitive to the drop rate. Please refer to Figure 14 in our paper for results under the same setting but with different drop rates. The test accuracy monotonically decreases as the defense rate increases along with the drop rate. Therefore, the appropriate value for the drop rate mainly depends on the relative importance of test accuracy and robustness. For example, if a task mainly requires high defense performance instead of high test accuracy, a large drop rate should be used.
>
> > I am upgrading my reviews after the rebuttal, which actually has convinced me that there is something interesting going on in this paper. However, I'm not entirely convinced as the approach seems to be ad hoc. the intuitions provided are somewhat satisfactory, but it's not clear why the method works.
>
> We have described our insights on why the proposed method works both in the paper and in our rebuttals which are supported by extensive experiments. Yet we believe that at present, the primary question for adversarial example research is to understand the prevalent existence of adversarial examples with small perturbations against various machine learning methods. Only when this primary question is well-understood can one further discuss why a new method is more robust. However, there is currently no satisfactory theory or insight on why adversarial examples exist. To make the question more specific, let us take the training data of CIFAR-10 as an example. A simple experiment shows that the average \ell_\infty distance between two images from different categories is larger than 100. Even the smallest \ell_\infty distance between two differently categorized images is 54, the half of which is significantly larger than 16, the common perturbation scale required to fool the model on CIFAR-10 by common attack methods. This fact demonstrates that the classifiers we currently use have a lot of room for improvement in terms of robustness. Random Mask is an attempt to improve robustness of common classifiers while maintaining generalization.

---

### Official Review · AnonReviewer1 · 2018-11-04
**Simple but efficient method to increasing the robustness of CNN against adversarial attacks**

**Rating:** 7
**Confidence:** 3

**Review:**

The authors propose a simple method for increasing the robustness of convolutional neural networks against adversarial examples. This method is simple but seems to achieve surprisingly good results. It consist in randomly remove neurons from the network architecture. The deleted neurons are selected before training and remain deleted during the training and test phase.  The authors also study the adversarial examples that still fool the network after applying their method and find than those examples also fool human. This finding raises the question of what is an adversarial example if both humans and networks are fooled by the same example.

Using Random Masks in neural network is not a new idea since it was already proposed for DropOut or DropConnect (Regularization of Neural Networks using DropConnect, ICML2013) and in the context of adversarial attacks (Dhillon et al. 2018)  as reported by the authors. The discussion (Section 2) about the impact of random masks on what convolution layers capture in the spatial organisation of the input is interesting: whereas standard CNNs focus on detecting the presence of a feature in the output, random mask could force the CNN layers to learn how a specific feature distributes on the whole input maps. This limitation of the CNN has already been pointed up and solutions have been proposed for example Capsule Networks (Dynamic Routing Between Capsules, NIPS 2017). This intuition is experimentally supported by a simple random shuffle by block of the input image  (Appendix A).

In Section 3, the authors present a large number of experiments to demonstrate the robustness of their method. Most of the details are given in the 13 (!) pages of appendix. Experiments against black-box attack, random noise, white-box attack, grey-box are presented. Most of the experiments are on CIFAR10 but one experiment is also presented on MNIST. One could regret that only one architecture of CNN is tested (ResNet18) except for gray-box attack, for which DenseNet121 and VG19 are tested. One could ask why the type of models tested is not consistent across the different experiments.  For black-box attack, random masks compare favourably to Madry’s defence. For white box defence, Random Mask is not compared to another defence method, which seems a weakness to me but I am not familiar enough with papers in this area to estimate if this is a common practice. In most of the experiments, the drop ratio is between 0.5 and 0.9, which seems to indicate that the size the initial network could be reduced by more than 50% to increase the robustness to attack. This ratio is larger than what is usually used for dropout (0.5 at most).

In section 3.3, different strategies for random masks are explored : where to apply random masks, random mask versus random channels, random masks versus same masks. Results are given in table 2. The caption of Table 2 could be more explicit : what are the presented percent ?

Experiments on masking shallow versus deep layers are interesting. Best results for robustness are obtained with masking shallow layers at quite a high ratio (0.9). One could ask if this result could be due to the type or the parameters of adversarial attacks which are not adapted to such a high sparseness on shallow layers or to the specific kind of sparseness induced by the masks. A comparison to a regular network with the same number of free parameters as the masked network could give insight on this aspect.

pros : simple to implement, good robustness shown agains a variety of attack types
cons : mainly tested on a single architecture (ResNet) and on a single datatbase CIFAR. Maybe not robust against the latest techniques of adversarial attack.

---

> ### Public Comment · (anonymous) · 2018-11-14
> **Very similar to Stochastic Activation Pruning**
>
> The proposed approach is very similar to Stochastic Activation Pruning at ICLR'18, which was one of the defenses shown to be broken by Athalye et al. 2018. Unfortunately the authors do not run the attacks that beat SAP on their model which makes it difficult to know if it will be effective.

---

> > ### Author Response · Authors · 2018-11-22
> > **Random Mask is essentially different from SAP**
> >
> > Thanks for your comment.
> >
> > In [1], SAP is applied to pretrained networks without fine tuning. This can be interpreted as randomly dropping some neurons and scaling up the rest at test time. Our approach, however, randomly masks out neurons before training and thus changes the network structure. Hence it is essentially different from dropout at test time. In terms of performance, SAP decreases test accuracy significantly if the percentage of sampled neurons is low, while our model preserves high test accuracy even if the drop ratio is 90%. Our model also has better black-box defense performance than SAP (See Figure 1 (c) SAP-100 vs Dense in [1]). You may compare the defense success rates of our model and those of SAP since the two models are both tested using perturbation scale of 8, 16 and 32 on CIFAR-10.
> >
> > [1] Dhillon, Guneet S., et al. "Stochastic activation pruning for robust adversarial defense." arXiv preprint arXiv:1803.01442 (2018).

---

> ### Author Response · Authors · 2018-11-22
> **Your suggestions have been accounted for in our revised version**
>
> Thanks for your review.
>
> > The caption of Table 2 could be more explicit : what are the presented percent?
>
> Thanks for your suggestion on the caption of Table 2. We have fixed it.
>
> > A comparison to a regular network with the same number of free parameters as the masked network could give insight on this aspect.
>
> We already compared the performance of our structure to that of a regular network with the same number of neurons in Section 3.3. The control group is called Channel Mask, which means randomly dropping out whole channels (or kernels, equivalently) according to the same ratio. The results (0.5-Shallow and 0.5-Shallow_{DC}) show that simply reducing the number of neurons without breaking the symmetry of channels cannot significantly enhance defense performance. We hope that the comparisons made in Section 3.3 and the full information on experiments presented in Appendix F.5 can bring about insights on how to improve the robustness of a network via changing its structure.
>
> > mainly tested on a single architecture (ResNet) and on a single database CIFAR.
>
> Thanks for your suggestion that Random Mask should be tested on architectures other than ResNet-18, and on datasets other than CIFAR-10. In the new version of our paper, we have added experiments on CIFAR-10 and MNIST with Random Mask applied to ResNet-50, DenseNet, SENet and VGG.
>
>
> > Maybe not robust against the latest techniques of adversarial attack.
>
> We have tested the robustness of CNNs with Random Mask with respect to black-box defense on three popular attack methods (FGSM, PGD and CW), and most of the results are listed in Appendix F.5. In particular, [1] suggested that PGD is “a ‘universal’ adversary among first-order approaches”. Besides, our model is able to effectively defend against Gaussian random noise and to generate human-fooling adversarial examples. As for your suggestion to test on more advanced black-box attacks, we think they are out of the scope of this work since these methods have few baselines to compare with. Most works concerning the robustness of neural networks focus on the three attack methods mentioned above. In our paper, the black-box defense mainly serves as an approach to evaluating robustness, and we believe the three attack methods we used are sufficient for this purpose.
>
> [1] Madry, Aleksander, et al. "Towards deep learning models resistant to adversarial attacks." arXiv preprint arXiv:1706.06083 (2017).

---

> > ### Public Comment · (anonymous) · 2018-12-11
> > **More powerful transfer-based black-box attacks**
> >
> > Most of the experiments are based on the transfer-based black-box setting. I think the author should not claim their method is robust against general black-box attacks since there are score-based [1] and decision-based [2] methods.
> >
> > However, as far as I'm concerned, PGD and CW attacks are popular methods in the white-box setting. Many transfer-based black-box attacks [3,4] are proposed and studied in the literature. There is an "overfitting" phenomenon [3] of the adversarial examples generated by PGD and CW. So changing the network architecture (e.g., masking neurons) could be useful to defend against them. But it's not clear whether the proposed method is generally robust to more powerful transfer-based black-box attacks.
> >
> > [1] Ilyas et al., Black-box Adversarial Attacks with Limited Queries and Information. ICML 2018.
> > [2] Brendel et al., Decision-based Adversarial Attacks. ICLR 2018.
> > [3] Dong et al., Boosting Adversarial Attacks with Momentum. CVPR 2018.
> > [4] Xie et al., Improving Transferability of Adversarial Examples with Input Diversity. Arxiv 2018.

---

> > > ### Author Response · Authors · 2018-12-12
> > > **Still effective against the mentioned transfer-based black-box attack**
> > >
> > > > There is an "overfitting" phenomenon [3] of the adversarial examples generated by PGD and CW. So changing the network architecture (e.g., masking neurons) could be useful to defend against them. But it's not clear whether the proposed method is generally robust to more powerful transfer-based black-box attacks.
> > >
> > > For your concern, we generate adversarial examples by the attack method proposed in [1] and use them to attack both normal and Random Masked network. The results are listed in the table below. Networks in the first row are the source models from which we generate adversarial examples by MI-FGSM ([1]). The results show that Random Mask is still effective against MI-FGSM ([1]).
> > >
> > > ---------------------------------------------------------------------
> > > |                              | DenseNet | SENet   | TestAcc |
> > > | Normal ResNet |    12.58%   |   8.44%  |    95.33  |
> > > | Random Mask   |    58.11%   | 50.81% |    93.39   |
> > > ---------------------------------------------------------------------
> > >
> > > > Most of the experiments are based on the transfer-based black-box setting. I think the author should not claim their method is robust against general black-box attacks since there are score-based [1] and decision-based [2] methods.
> > >
> > > Please refer to the last paragraph of our reply to AnonReviewer1.
> > >
> > > [1] Dong et al., Boosting Adversarial Attacks with Momentum. CVPR 2018.

---

> > > > ### Public Comment · (anonymous) · 2018-12-19
> > > > **Thanks**
> > > >
> > > > Thanks for your reply!

---

### Official Review · AnonReviewer2 · 2018-11-06
**interesting observations; but what insights to get out of it?**

**Rating:** 6
**Confidence:** 3

**Review:**

This paper proposes a surprisingly simple technique for improving the robustness of neural networks against black-box attacks. The proposed method creates a *fixed* random mask to zero out lower layer activations during training and test. Extensive experiments show that the proposed method without adversarial training is competitive with a state-of-the-art defense method under blackbox attacks.

Pros:
 -- simplicity and effectiveness of the method
 -- extensive experimental results under different settings

Cons:
 -- it's not clear why the method works besides some not-yet-validated hypotheses.
 -- graybox results seem to suggest that the effectiveness of the method is due to the baseline CNNs and the proposed CNNs learning very different functions; source models within the same family still produce strong transferable attacks. It would have been much more impressive if different randomness could result in very different functions, leading to strong defense in the graybox setting.

---

> ### Comment · AnonReviewer2 · 2018-11-06
> **Lack of SOTA black box defenses**
>
> Besides, it concerns me that the paper didn't make a comparison with SOTA defenses besides Madry et al.
>
> Since Athalye et al. (2018) only invalidates other defense methods in the white-box setting, some of them could still be robust in the black-box setting, I assume?

---

> > ### Author Response · Authors · 2018-11-22
> > **SOTA performance**
> >
> > Thanks for your suggestion. Among the other pieces of work mentioned in [1], there is indeed one work [3] which combines two techniques, adversarial training and thermometer encoding, to achieve better defense performance than [2].
> >
> > 1.We conducted experiments comparing Random Mask with [3] and found that our method is more robust. Please see the table attached below.
> >
> > 2.We would like to emphasize that Random Mask is NOT a defense method. A network with Random Mask is an architecture that is designed to be as robust as possible by itself, without using any defense method. Of course, Random Mask can be combined with existing defense methods, for example adversarial training ([2]) which we compared with in the paper, to achieve even better results.
> >
> > -----------------------------------------------------------------------
> > |                                      |  FGSM   |   PGD  | TestAcc |
> > | Random Mask           |  91.31    |   93.67 |   91.86   |
> > -----------------------------------------------------------------------
> > |                                      |      DefenseRate  | TestAcc |
> > |Thermometer(16) [3]|           88.25          |   89.88   |
> > |Thermometer(32) [3]|           86.06          |   90.30   |
> > -----------------------------------------------------------------------
> >
> > [3] uses both thermometer encoding and adversarial training on a Wide ResNet with a width factor of 4. For comparison, we tested on the same network structure with Random Mask applied to the shallow layers of it. The performance data of [3] are found in Table 12 in [3]. However, what attack methods were used to obtain the defense rates shown in Table 12 is not clear. There is a contradiction between the claimed method (PGD) and the method in [2] (FGSM) which they compared to in Table 12. Therefore, we listed the defense rates of our model against both attack methods.
> >
> > [1]Athalye, Anish, Nicholas Carlini, and David Wagner. "Obfuscated gradients give a false sense of security: Circumventing defenses to adversarial examples." arXiv preprint arXiv:1802.00420 (2018).
> > [2]Madry, Aleksander, et al. "Towards deep learning models resistant to adversarial attacks." arXiv preprint arXiv:1706.06083 (2017).
> > [3]Buckman, Jacob, et al. "Thermometer encoding: One hot way to resist adversarial examples." (2018).

---

> ### Author Response · Authors · 2018-11-22
> **Insights of the Random Mask**
>
> Thanks for your review.
>
> > it's not clear why the method works besides some not-yet-validated hypotheses.
>
> Although our method seems simple, it is carefully designed. The method removes some neurons in the *shallow* layers of the neural network. We emphasize that the neurons in the shallow layers have relatively small receptive fields. Therefore, by removing a part of such neurons, the remaining neurons in the shallow layers not only response to features, but also automatically record the locations of the features. As a result, these remaining neurons in shallow layers together detect the *spatial structure* of the features, much better than the standard neural networks.
>
> The motivation of our design comes from the recent observation ([1]) of adversarial examples. In many cases, the adversarial examples change a patch of the original image so that the perturbed patch looks like a small part of the incorrectly classified object. This perturbed patch, although contains crucial features of the incorrectly classified object, usually appears at the wrong location and does not have the right spatial structure with other parts of the image. For example, the adversarial example of a panda image is misclassified as a monkey because a patch of the panda skin is perturbed adversarially so that it alone looks like the monkey’s face. However, this patch does not form a right structure of a monkey with other parts of the images (see Figure 11 in [1]).
>
> In sum, current deep neural networks are strong at detecting features, but relatively weak at telling if the spatial location/structure is right. Random Mask tries to strengthen the ability of neural networks in utilizing the spatial information.
>
> > graybox results seem to suggest that the effectiveness of the method is due to the baseline...
>
> The grey-box attacks are very similar to white-box attacks in our setting. We demonstrate in the paper that the adversarial examples (eps = 16,32)  generated by the white-box attacks for Random Mask often fool human as well. We then raise the following questions: 1) Should these adversarial examples be classified as their original categories? 2) How to evaluate the robustness of a method? 3) Can we entirely rely on the currently used performance measures?
>
> In sum, one of the major goal of this paper is to move a tiny step towards a better understanding of the problem of adversarial example.
>
> [1] Liu, Mengchen, et al. "Analyzing the Noise Robustness of Deep Neural Networks." arXiv preprint arXiv:1810.03913(2018).

---

### Public Comment · (anonymous) · 2018-10-02
**Minor suggestion for revision**

This sentence could be misconstrued as implying that adversarial examples are specific to deep neural networks: "Despite the great success in numerous applications, recent studies have found that deep CNNs are vulnerable to some well-designed input samples named as Adversarial Examples". It would be better to rewrite this to say something like "all known machine learning models including deep CNNs". I don't intend for this piece of feedback to influence the reviewers toward accepting or rejecting the paper, it is just a suggestion for revision to slightly improve clarity.

---

> ### Author Response · Authors · 2018-11-22
> **Thanks for your suggestion**
>
> Since we are not trying to define “adversarial examples” in that sentence, we think using the expression “named as” is fine. We did not mention the existence of adversarial examples in other machine learning models because our paper mainly focus on CNNs. Further discussion is certainly welcomed if you have other suggestions.

---

### Public Comment · (anonymous) · 2018-10-02
**Minor clarification question**

Is Random Mask intended to be specific to convolutional neural networks? It seems like if you applied Random Mask to a fully connected layer, it would be equivalent to just using a smaller layer. I don't intend for this question to affect whether the paper is accepted or rejected, just asking to further my own understanding. I'm sorry if this is already explained in the paper and I've missed it.

---

> ### Author Response · Authors · 2018-11-22
> **Based on convolutional network but potential for generalizing**
>
> Thanks for your comment. Please see Section 2 for the intuition of Random Mask. It is indeed based on a convolutional network structure. Nonetheless, it is definitely worth trying to apply Random Mask or some similar ideas to network structures other than CNNs, and we will surely be pleased if more works concerning Random Mask come out in the future.

---

### Public Comment · (anonymous) · 2018-10-02
**Minor clarification question**

Which attack is used in Figure 7? For example, is it a black box attack? If so, which model are the examples transferred from? What is the size of the max norm constraint?

---

> ### Author Response · Authors · 2018-11-22
> **Also under Madry’s setting**
>
> Thanks for your reply. In our revision, Figure 7 is changed to Figure 14. It serves as a complement of Table 1 in the main body, and the attack also follows [1]’s setting which is mentioned in the caption of Table 1 and is elaborated in Appendix F.1.
>
> [1]Madry, Aleksander, et al. "Towards deep learning models resistant to adversarial attacks." arXiv preprint arXiv:1706.06083 (2017).

---

### Public Comment · (anonymous) · 2018-10-02
**Are there any experiments with small perturbations?**

Several images in this paper show that with large perturbations, humans also change their output class. Does this paper also evaluate with small perturbations, that are more likely to be class-preserving? For example, Madry et al use epsilon=8, and it would be nice to compare directly to this threat model. Sorry if this is already in the paper and I've missed it.

---

> ### Public Comment · (anonymous) · 2018-10-03
> **same question**
>
> The paper said the scale is 0.3x255 (caption of Tab.6), which I think is pretty large. The images are hardly intelligible.
>
> According to previous works, a scale smaller than 0.063 (16/255) is reasonable.

---

> > ### Public Comment · (anonymous) · 2018-10-03
> > **Tab.6 is for MNIST**
> >
> > It seems that Tab.6 is for MNIST where small perturbations are no use since the data can be easily binarized. For datasets like Cifar, 16/255 is certainly reasonable in my experience. I'm sorry if I miss something you mentioned and I'm pleasure to have further discussion.

---

> > > ### Public Comment · (anonymous) · 2018-10-03
> > > **On Cifar10, 8/255 makes more sense**
> > >
> > > I think 8/255 is a standard because the state-of-the-defense, i.e., PGD adversarial training, achieves 44.71% under DAA when the perturbation is 8/255. (under white-box adaptive setting)
> > > I haven't seen a work that can really outperform PGD adversarial training until now. You can refer to the "obfuscated gradients" paper https://arxiv.org/abs/1802.00420 and its Github page https://github.com/anishathalye/obfuscated-gradients.

---

> > > > ### Author Response · Authors · 2018-11-22
> > > > **Please refer to our results**
> > > >
> > > > Thanks for your comment. Actually our paper included experiments with perturbation scale 8, which should address your concern. Please refer to Section 3.1.1 and Appendix F.1, F.5..

---

> ### Author Response · Authors · 2018-11-22
> **Experimental results are under several common settings**
>
> Thanks for your replies and discussions. We think that there might be some misunderstanding here. The presented adversarial examples from the CIFAR-10 dataset are generated using perturbation scale ε=16, 32, yet we have evaluated our model in terms of defense success rate using different ε values ranging from 4 to 32 - especially the results under the setting of [1] are presented in Section 3.1.1. We have also observed adversarial examples generated using different ε values. In fact, when ε=8, most adversarial examples from CIFAR-10 against our model are hard for humans to classify. The reason why we present images after larger perturbations is that we found examples capable of “fooling” human eyes using ε values larger than 8 (See Figure 1). Moreover, as is mentioned in one of your replies, 16/255 is reasonable for ordinary networks. However, in contrast to our model, adversarial examples generated against *normal* CNNs with ε=16 are similar to the original images added with some noise which can be easily ignored by humans.
>
> Besides, as is also mentioned in one of your replies, Table 5 (which extends Table 6 in the original version) is for MNIST, and the perturbation scale used there should not be compared with that used in experiments on CIFAR-10.
>
> [1]Madry, Aleksander, et al. "Towards deep learning models resistant to adversarial attacks." arXiv preprint arXiv:1706.06083 (2017).

---

### Public Comment · (anonymous) · 2018-10-02
**I like appendix G.3**

I like that you test using an attack based on transfer from one model  that was trained with Random Mask to another model that also has Random Mask. This helps to show that the defense works even if the attacker knows you are using Random Mask, i.e. that the defense hasn't just made the model *different* from a normal CNN.

---

### Public Comment · (anonymous) · 2018-10-03
**Interesting but few questions**

1. Why does random mask improve the network robustness against adversarial samples? The reason is not intuitive to me, is there any precise theoretical support？

2. Many defenses can work under simple black settings, so what is the advantage of your framework? Besides, there are some advanced black-box settings, e.g., 1. using certain inputs and the outputs of the model to train a similar model as the target 2. attack an ensemble of models as in https://arxiv.org/pdf/1710.06081.pdf  3. GAN-based black-box attack as in https://arxiv.org/abs/1801.02610
Since you claim your defense achieves state-of-the-art performance against black-box adversarial attacks, have you tested those advanced black-box settings?

---

> ### Author Response · Authors · 2018-11-22
> **Heuristic but meaningful results**
>
> Thanks for your reply.
>
> The theoretical analysis of Random Mask is highly dependent on the theory of CNN which is not clear yet. Therefore, we only show some of our intuition (See Section 2), and we used the Random Shuffle experiment (See Appendix A) to verify our intuition that our model learns more “reasonable” features than a normal CNN. A brief restatement of our insights can be found in our reply to AnonReviewer2. Also, we regard the theoretical analysis of Random Mask as an interesting future work and further discussion is welcomed.
>
> We have mentioned in the paper that Random Mask is an efficient method with simple implementation while enhances robustness significantly. As for your concern on the black-box attack methods, you may refer to the last paragraph of our reply to AnonReviewer1.

---

### Public Comment · (anonymous) · 2018-10-31
**Lack of SOTA black box attacks**

Despite making claims about being robust to black-box attacks, this paper does not seem to actually perform any state of the art black-box attacks. See for example

Decision Based Adversarial Attacks ICLR'18
Adversarial Risk and the Dangers of Evaluating Against Weak Attacks ICML'18
Black-box Adversarial Attacks with Limited Queries and Information ICML'18

---

> ### Author Response · Authors · 2018-11-22
> **Thanks for your comment**
>
> Please refer to the last paragraph of our reply to AnonReviewer1.

---

### Author Response · Authors · 2018-11-22
**Revision uploaded**

Here is a summary of the revision:
1. We provide a detailed explanation of why Random Mask is robust in Section 2.
2. We provide more results of Random Mask applied to different network structures (ResNet-50, DenseNet-121, SENet-18, VGG-19) on CIFAR-10 and MNIST datasets in Appendix F.2. These results are consistent with our original version on ResNet-18.

---

### Meta-Review · Area_Chair1 · 2018-12-15
**Some interesting ideas but is not mature enough for publication**

**Confidence:** 5
**Recommendation:** Reject

**Metareview:**

This paper presents a new technique for modifying neural network structure, and suggest that this structure provides improved robustness to black-box attacks, as compared to standard architectures. The paper is very thorough in its experimentation, and the method is simple and quite easy to understand. It also raises some important questions about adversarial examples.

However, there are serious concerns regarding the evaluation methodology. In particular, the authors claim "black-box robustness" but do not test against any query-based attacks, which are known to perform better against gradient masking-based adversarial defenses. Furthermore, it is not clear why one would expect adversarial examples to transfer between models representing two completely different functions (i.e. from a standard model to a random mask model). So, the gray-box evaluation is much more informative and, unfortunately, random-mask seems to provide little to no robustness in this setting.

Given how fundamental sound and convincing evaluation is for proposed defense methods, the submission is not ready for publication yet. In particular, the authors are urged to (a) evaluate on stronger black-box attacks, and (b) compare to a baseline that is known to be non-robust, (e.g. JPEG encoding or SAP), to verify that these results are actually due to black-box robustness and not simply obfuscation.